



**Role of time-averaging of eddy covariance fluxes on water use efficiency**
**dynamics of Maize crop**
Arun Rao Karimindla, Shweta Kumari, Saipriya SR, Syam Chintala*, and BVN Phanindra
Kambhammettu
Department of Civil Engineering, Indian Institute of Technology Hyderabad, Telangana, India.
*Corresponding author: E-Mail: ce22resch11012@iith.ac.in; Tel: +91 7997014429
**Abstract**
Direct measurement of carbon and water fluxes at high frequency makes eddy
covariance (EC) as the most preferred technique to characterize water use efficiency (WUE).
However, reliability of EC fluxes is hinged on energy balance ratio (EBR) and inclusion of
low-frequency fluxes. This study is aimed at investigating the role of averaging period to
represent EC fluxes and its propagation into WUE dynamics. Carbon and water fluxes were
monitored in a drip-irrigated Maize field at 10 Hz frequency and are averaged over 1, 5, 10,
15, 30, 45, 60, and 120 minutes considering daytime unstable conditions. Optimal averaging
period to simulate WUE fluxes for each growth stage is obtained by considering cumulative
frequency (ogive) curves. A clear departure of EBR from unity was observed during dough
stage of the crop due to ignorance of canopy heat storage. Error in representing water (carbon)
fluxes relative to the conventional 30 min. average is within $\pm$ 3 % ($\pm$ 10 %) for 10-120 min.
averaging and is beyond $\pm$ 3 % ($\pm$ 10 %) for other time-averages. Ogive plots conclude that
optimal averaging period to represent carbon and water fluxes is 15-30 min. for 6th leaf and
silking stages, and is 45-60 min. for dough and maturity stages. Dynamics of WUE considering
optimal averaging periods are in the range of 1.49 $\pm$ 0.95, 1.37 $\pm$ 0.74, 1.39 $\pm$ 0.79, and 3.06 $\pm$
0.69 $\mu$mol mmol$^{-1}$ for the 6th leaf, silking, dough, and maturity stages respectively. Error in
representing WUE with conventional 30 min. averaging is marginal (< 1.5 %) except during
the dough stage (12.12 %). Our findings can help in developing efficient water management
strategies by accurately characterizing WUE fluxes from the EC measurements.
**Keywords:** Eddy covariance, Maize crop, Time-average, Energy balance ratio, Ogive
function, Water use efficiency.
**Research Highlights:**


1. The time-averages that yield the most effective energy balance closure are identified as 45 and 60 minutes.
2. Insufficiently short time-averages such as 1 and 5 minutes, as well as excessively long-time-averages such as 120 minutes, resulted in a high relative error in representing carbon and water fluxes.
3. The conventional 30-minute averaging period proved to be insufficient in capturing low-frequency fluxes, necessitating the use of longer averaging periods.
4. Time-averaging of Eddy Covariance fluxes needs to be performed in accordance with crop growth stage.

## 1.0 INTRODUCTION

Water use efficiency (WUE) is an important eco-hydrologic trait relating two important processes of plant metabolism namely carbon fixation (via photosynthesis) and water consumption (via transpiration) (Bramley, 2013). The need for achieving food security with diminishing water resources under changing climate has made WUE as the controlling parameter in planning and design of irrigation strategies (Tang, 2015). Depending on the scale of investigation, WUE can be quantified at: i) leaf, ii) plant, iii) ecosystem, or iv) regional scales (Medrano, 2015). Of these, ecosystem WUE has taken precedence in irrigation and agronomy due to: i) accurate and reliable measurement using micrometeorological techniques, ii) ability to evaluate the role of various water conservation techniques on ecosystem productivity, and iii) understand the relation between carbon and water cycles in response to changes in climate (Tang, 2015; Tong, 2014).

Eddy covariance (EC) is a non-destructive, micrometeorological technique for direct measurement of water vapour ($H_2O$) and carbon ($CO_2$) fluxes between vegetation and atmosphere at high temporal frequency (Aubinet, 1999; Leclerc and Foken, 2014). EC method precisely measures the overall transfer of heat, mass, and momentum between the earth's surface (such as vegetation) and the atmosphere. This is achieved by estimating the covariance of turbulent fluctuations in vertical wind (referred to as eddies) with respect to the specific flux under consideration such as $H_2O$, $CO_2$, temperature. EC represents the scalar fluxes of interest (representative of eco-hydrological processes) from a region upwind of the measurement known as the footprint. At ecosystem scale, WUE is estimated as the ratio of net primary product (NPP): proxy for photosynthesis to evapotranspiration (ET): proxy for water



consumption (Peddinti, 2020). WUE is a key eco-hydrologic trait that is used to analyse the
role of climate change, drought, deficit irrigation, and management strategies on ecosystem
productivity. Currently, EC is the most accurate and reliable method for estimating carbon and
water fluxes, hence WUE at ecosystem scale (Tong, 2009). A number of studies have
demonstrated the efficacy of EC in estimating WUE across a wide range of ecosystems (Tang,
2015; Tong, 2014; Wang, 2017). Despite improvements in measurement accuracy, data
sampling, and processing techniques, EC method suffers from the drawback of lack of
conservation among the energy terms, resulting in energy balance closure (EBC) problem
(Charuchittipan, 2014; Foken, 2011; Reed, 2018). Lack of EBC as observed in EC system is
reported across diverse ecosystems ranging from simple bare soils (Oncley, 2007), to
homogeneous grasslands (Twine, 2000), to heterogeneous croplands (Peddinti and
Kambhammettu 2019), to complex forest (Charuchittipan, 2014; Wilson, 2002). Apart from
the errors associated with instrumentation, measurement, and neglected energy sinks, lack of
EBC at the EC sites is also attributed to the omission of low frequency secondary circulations
in the turbulent flux estimation (Wilson, 2002). This problem can be circumvented by choosing
appropriate averaging period during flux estimation, the selection of which is based on: i)
'ensemble block time-averaging method' (Finnigan, 2003; Malhi, 2004; Sakai, 2001), and ii)
'ogive method' (Berger, 2001).
A number of studies have highlighted the importance of averaging period in quantifying
the EC fluxes, with an objective to obtain optimal time-averaging period under various canopy
and surface roughness conditions. While smaller averaging periods (15-30 min.) are suitable
for managed croplands, flux estimation from forest and tall canopies demand longer averaging
periods (60-120 min.) due to the presence of large-sized, slow moving eddies (Finnigan, 2003;
Sakai, 2001; Sun, 2006). Zhang (2013) concluded that time-averaging of EC fluxes has to be
done in accordance with the observation scale. In an analysis of Chengliu riparian forest in
China, they found that lower time-averaging periods (15 min.) are suitable for daily variation
of EC fluxes, whereas higher time-averaging periods (60 min.) are suitable for long-term flux
computations. A similar observation was made by Lee (2004) over farmlands. In a wheat field
in Yucheng, China, 10 min. and 30 min. averaging periods were found suitable for diurnal and
long-term flux observations respectively. Flux observations over a Maize crop at Daxing
experimental station in China conclude that optimal time-averaging period has to be considered
in accordance with crop growth stage (Feng, 2017). However, they observed a marginal (< 3



%) error in representing the fluxes at conventional 30 min. averaging relative to the optimal
averaging obtained for each growth stage.
Maize is the third most important cereal crop in India after rice and wheat, and accounts
for about 10 % of total food production in the country (Sharma, 2018; Ficci 2014). Inspite of a
huge area under cultivation (9.4 MHa), high production (23 million tons), and enormous water
consumption (18 BCM), both crop productivity (2.5 t ha$^{-1}$) and crop water productivity: CWP
(1.83 kg m$^{-3}$) of Indian Maize are far lower than corresponding world averages (Sharma, 2018).
Low CWP (hence, WUE) of Indian Maize can be attributed to: i) a high dependence (85 %) on
erratic, uncertain rainfall, ii) low adoption of hybrid varieties, iii) improper drainage facilities
leading to water logging, and iv) unscientific application of irrigation water without analysing
soil-water-crop interactions (Shankar, 2012). Thus, an accurate quantification of WUE and its
temporal variation during the crop cycle is essential for effective irrigation water management
of Maize crop (Medrano, 2015).
While the effect of time-averaging on carbon and water fluxes measured at EC sites is
reported, the effect on their interaction term, i.e. WUE, which is crucial in irrigation water
management is unexplored. Evaluation of time-averaging period on WUE dynamics is
necessary to understand the contribution of low and high frequency photosynthetic carbon and
evaporative water fluxes generated from various field management strategies. Also, most of
the EC flux studies are confined to data rich AmeriFLUX, EuroFLUX, and ChinaFLUX sites,
with limited focus to Indian fragmented croplands. This motivates the present study, and the
objectives of this study are as follows: i) investigate the role of time-averaging of EC fluxes on
EBR and WUE dynamics, ii) compute optimal averaging period to simulate carbon and water
(hence, WUE) fluxes of Maize crop, and iii) investigate the association of carbon, water, and
WUE fluxes between multiple averaging periods. Results of this study can help in designing
efficient management strategies using EC datasets in response to changes in WUE during the
crop cycle.

**2.0 MATERIALS AND METHODOLOGY**
**2.1 Site Description and Instrumentation**
Controlled Maize plots situated at Professor Jaya Shankar Telangana State Agricultural
University (PJTSAU), Hyderabad, Telangana, India (17°19′17″ N, 78°24′35″ E, 559 m above





sea level) forms the study area. The region is composed of red gravel to sandy loam soils with field capacity and wilting point in the ranges of 17.92 - 19.56 % and 8.2 – 9.87% respectively. As per Koppen-Geiger's classification, the region falls under tropical savanna climate zone (Aw) characterized by long dry and short wet seasons (Kottek, 2006). Mean annual precipitation of the region is 900 mm (IMD, 2019) with more than 80% occurring during the monsoon months (Jun-Sep). Temperatures are high during summer (38.33 ± 2.12 $^{0}$C) and low during winter (30 ± 2.20 $^{0}$C) months. Humidity of the region varies from 35% in summer to 73% in monsoon (CGWB, 2013). Mean seasonal wind speed is in the range of 1.5 to 2.7 m/s (Peddinti and Kambhammettu 2019). Hydro-geologically, the study area forms part of the Deccan plateau characterized by multiple layers of solidified flood basalt resulting from volcanic eruptions. Depth to groundwater ranges from 12 m (pre-monsoon) to 6 m (post-monsoon) (CGWB, 2013).

Meteorological parameters and turbulent fluxes were obtained for one crop season, i.e. 26 May to 06 Sep, 2019 using an open path eddy covariance (EC) flux tower. The flux system is composed of a 3D sonic anemometer (CSAT3, Campbell Sci. Inc., USA), and an open-path fast response infrared gas analyzer (IRGASON-EB-IC, Campbell Sci. Inc., USA) to measure $CO_2$ and $H_2O$ fluxes at 3 m above the canopy. Raw data was collected with a logger (CR1000, Campbell Sci. Inc., USA) at 10 Hz frequency. Additionally, slow response meteorological variables including precipitation (TE525-L-PTL, Tipping Bucket, Campbell Sci. Inc., USA), soil heat flux (HFP01SC-L-PTL, Campbell Sci. Inc., USA), solar radiation (CNR 4, Campbell Sci. Inc., USA), and soil moisture (CS616-L-PT-L, Campbell Sci. Inc., USA) were obtained at 10 min. intervals.

## 2.2 Data Collection and Processing

Table 1 shows the phenological stages of the Maize crop in the study area (Soujanya, 2021). Additionally, leaf-area index (LAI) and mean plant height were measured during the crop cycle (Table 1). The LAI was measured using the plant canopy analyser, whereas the plant height was measured using a ruler from the base of the plant to its crown. Maize crops of the experimental fields are sown on 25[th] May 2019 and harvested on 6[th] September 2019 with a base period of 104 days.

**Table 1:** Phenological growth stages and physical properties of the Maize crop



| S. No. | Growth stage | Start date | End date | Period Length (days) | Leaf Area Index ($m^2m^{-2}$) | Plant height (cm) |
|---|---|---|---|---|---|---|
| 1 | 6th leaf | 26/05/2019 | 12/06/2019 | 18 | 0.61 | 46.8 |
| 2 | Silking | 13/06/2019 | 19/07/2019 | 37 | 1.56 | 75.2 |
| 3 | Dough | 20/07/2019 | 12/08/2019 | 24 | 3.46 | 133 |
| 4 | Maturity | 13/08/2019 | 06/09/2019 | 25 | 3.03 | 134 |


157  Data from the EC system at 10 Hz frequency was converted to ASCII format using
LoggerNet (4.3) software (Campbell Scientific Inc., Logan, Utah, USA), and further
aggregated to various averaging periods (1, 5, 10, 15, 30, 45, 60, and 120 minutes). Post data
processing was done using EddyPro post-processing software (version 7.0.8, LI-COR, USA).
Primary corrections performed on the raw data include tilt corrections, turbulent fluctuations,
density fluctuations, frequency corrections and quality checks. Tilt corrections were made by
the double axis rotation method. The block average method and linear detrending method were
used to correct the turbulent fluctuations. Density fluctuation corrections were done using
Webb–Pearman–Leuning (WPL) method. Quality checks were performed following a flagging
policy proposed by Mauder and Foken (2006) (0-1-2 system). Flag set to "0" corresponds to
the best quality fluxes, "1" corresponds to fluxes acceptable for general analysis, and "2"
corresponds to poor quality fluxes that should be removed from the dataset. The resulting fluxes
may exhibit spikes, discontinuity, randomness etc. There is a need to perform secondary
corrections on the data that include flux spike removal (Vickers and Mahrt 1997), friction
velocity corrections, gap filling and uncertainty analysis (Finkelstein, 2001), skewness &
kurtosis removal, spectral corrections, and frequency corrections. To correct flux estimates for
low and high frequency losses due to instrument setup, intrinsic sampling limits of the devices,
and various data processing decisions, spectral corrections are performed. Additionally, slow
sensor meteorological data obtained at 1 min. interval were processed for different time-
averaging periods using the EddyPro post-processing software (version 7.0.8, LI-COR, USA).



### 2.3 Effect of time-averaging on EBR and EC fluxes

Lack of conservation among the measured energy terms of the EC tower is referred as energy balance closure (EBC). The available energy ($R_n$-G) is generally higher than the turbulent fluxes (H+LE), resulting in a positive balance (Eshonkulov, 2019) where $R_n$, G, H and LE correspond to net radiation, soil heat flux, sensible heat and latent heat respectively. Apart from instrument and measurement issues, this lack of energy closure is thought to be partly from averaging periods and coordinate systems (Finnigan, 2003; Finnigan, 2004; Gerken, 2018). The energy closure fraction, commonly termed as energy balance ratio (EBR) is used to evaluate the quality of EC data by examining energy fluxes at the surface (Chen and Li 2012), given by:

$$EBR = \frac{H+LE}{R_n-G} \tag{1}$$

EBR helps to determine the averaging period required to calculate H and LE fluxes over a range of landscapes (Chen and Li 2012). A high EBR (EBR $\geq$ 0.7) ensures reliability of EC observations for use with flux estimation.

Eddy fluxes are computed as the covariance between instantaneous deviation in vertical wind speed ($w'$) and scalar component of interest ($s'$) from their respective means, given by

$$\text{F} \approx \overline{\rho_a w' s'} \tag{2}$$

where $\overline{\rho_a}$ is the mean air density, and the overbar represents the time-average of eddy fluxes, which is of interest in the present study. Depending on the scalar component considered (ex: temperature, water vapour ($H_2O$), carbon dioxide ($CO_2$) concentration), corresponding eddy fluxes (ex: sensible heat, latent heat, carbon flux) are computed as below.

$$F_{CO_2} \approx \overline{\rho_a w' CO_2'} \tag{3}$$

$$F_{H_2O} \approx \overline{\rho_a w' H_2O'} \tag{4}$$

Ecosystem WUE is then estimated as the ratio of daytime carbon (net primary product) to water fluxes (evapotranspiration), observed during daytime unstable atmospheric conditions (08:00 am to 04:00 pm) given by:





$$WUE = \frac{NPP}{ET} = \frac{F_{CO_2}}{F_{H_2O}} \tag{5}$$
Fluxes originating from real-world sites are composed of both high frequency (turbulence) and
low frequency (advection) fluctuations, with a spectral gap in between. Isolating local
turbulence component for use with flux studies is achieved by choosing an appropriate
averaging period, $T_1$ (typically 30 minutes) on fast response measurements operating at high
frequency $T_2$ (Manon and Kristian 2020). Optimal averaging period ($T_1$) should be long enough
to reduce random error (Berger, 2001) and short enough to avoid non-stationarity associated
with advection (Foken & Wichura, 1996). The flux estimates (eq. 2) are further decomposed
into frequency dependent contributions, known as co-spectra $Co_{ws}$ (f) between vertical wind
velocity (w) and scalar of interest (s) for frequencies '$f$' (Manon and Kristian 2020). For an
accurate estimation of the flux, it is essential that the EC method is applied under conditions
where the flow is stationary, and all eddies carrying flux are sampled. Given that the flow
remains stationary, an 'ogive' serves as a check for the essential requirement to sample all
scales carrying the flux. Ogive function is well proposed to check if all low frequency fluxes
are included in the turbulent flux measured with the EC method (Foken & Wichura, 1996;
Foken et al., 2005). It is used to investigate the energy balance losses caused by low frequency
fluxes. Ogive analysis is performed to investigate the flux contribution from each frequency
range and to arrive at most suitable averaging period to capture most of the turbulent fluxes
(Desjardans, 1989; Charuchittipan, 2014). Ogive function thus provides the cumulative sum of
co-spectral energy starting from the highest frequency, given by:
$$Og_{ws}(f_0) = \int_{f_0}^{\infty} Co_{ws}(f) df \tag{6}$$
The point of convergence on the Ogive plot to an asymptote corresponds to optimal averaging
period ($T_1$) for use with averaging of high frequency turbulence fluxes. A total of eight
averaging periods, i.e., 1, 5, 10, 15, 30, 45, 60, and 120 minutes were considered to investigate
the role of time-averaging on EBR and EC fluxes, and further to arrive at the optimum
averaging period for use with WUE estimation. The biophysical and physiological
characteristics such as plant height, crop water requirement, LAI, etc. changes with respect to
the crop growth stage (Chintala et al., 2024) and have a significant effect on the EC fluxes. For
this reason, time-averaging of EC fluxes is separated based on crop growth stage.



### 2.4 Performance Evaluation


The ability of various averaging periods to close the energy balance and compute the
EC fluxes is evaluated using three goodness of fit indicators, namely: a) coefficient of
determination ($R^2$), b) root mean squared error (RMSE), and c) relative error (RE). While $R^2$
and RMSE aim to quantify the error in closing the energy balance, RE is aimed to compute the
error in estimating EC fluxes with conventional 30 min. averaging period relative to optimal
averaging period.
Root mean square error (RMSE) measures overall accuracy in closing the energy balance for
a given averaging period by penalizing large errors heavily, given by:
$$RMSE = \left[\frac{\sum_{i=1}^{n}(R_n - G)_i - (H + LE)_i}{n}\right]^{0.5} \tag{7}$$

Where n is the number of observations.
Coefficient of determination ($R^2$) is a measure of the strength of linear association between
turbulent fluxes and available energy, given by:
$$R^2 = \left\{\frac{\sum_{i=1}^{n}\left[(R_n - G)_i - \overline{(R_n - G)}\right]^2\left[(H + LE)_i - \overline{(H + LE)}\right]^2}{\sqrt{\sum\left[(R_n - G)_i - \overline{(R_n - G)}\right]^2\left[(H + LE)_i - \overline{(H + LE)}\right]^2}}\right\}^2 \tag{8}$$

Relative error (RE) provides the disparity in the fluxes estimated with conventional (30 min.)
relative to the fluxes estimated with optimal averaging period, given by:
$$RE = \left[\frac{\{F_{opt} - F_{30min}\}}{F_{opt}}\right] \times 100 \tag{9}$$

where $F_{opt}$ and $F_{30}$ are the flux of interest considering optimal and conventional (30 min.)
averaging periods.
A high $R^2$ (close to 1), low RMSE (close to zero), and low RE (close to zero) is considered to
be the optimal choice in representing the EC fluxes.

### 3.0 RESULTS AND DISCUSSION


### 3.1 Diurnal variations in energy balance components


To understand the energy variation in response to rapid changes in meteorological
conditions, we analysed the diurnal variations in energy balance components. Figure 1 shows





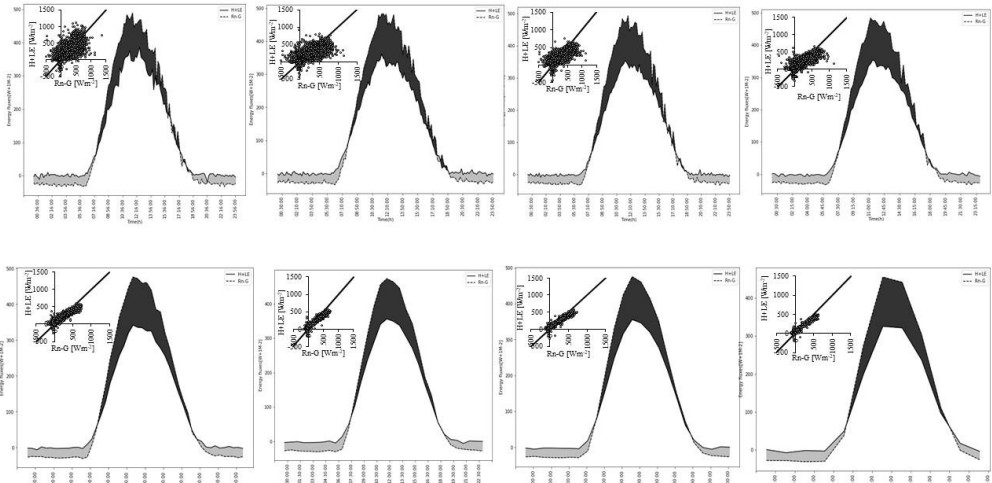


**Figure 1:** Diurnal variations in energy balance components (available energy: $R_n$-G and turbulent fluxes: H+LE) during the crop cycle with different averaging periods. Inset: Scatter-plots between the two datasets.

the diurnal variations in available energy ($R_n$-G) and turbulent fluxes (H+LE) averaged over the crop cycle for various time-averages. The diurnal variations of ($R_n$-G) and (H+LE) are bell-shaped, with peak occurring at around noon ($480.16 \pm 14.15$ Wm$^{-2}$, $356.23 \pm 18.51$ Wm$^{-2}$) (Figure 1). The energy balance difference (shaded areas of the figure) is found to be positive ($76.88 \pm 43.14$ Wm$^{-2}$) during daylight hours (08:00 am to 06:00 pm) and is negative ($-24 \pm 11.65$ Wm$^{-2}$) for the remaining time. The vertical offset between the two curves, representing the residual of energy balance is highest around the noon ($142.39 \pm 19.42$ Wm$^{-2}$), and is consistent between the averaging periods. For an average site-day, the cumulative energy balance difference was found to be constant with a mean of $1811 \pm 91.56$ Wm$^{-2}$ at all averaging periods. The cumulative energy balance difference is crossing the 'zero' line at around 11:30 am. The variation is rough at lower averaging periods due to a high sample size (n= 10859 at T = 1 min.) and is gradually smoothened towards higher averaging periods (n= 811 at T = 120 min.). The slope of regression lines between (H+LE) and ($R_n$-G) considering all averaging periods are in the range of 0.59 to 0.71 with a mean of $0.65 \pm 0.041$. The intercept is ranged from 19.01 to 31.56 Wm$^{-2}$. The best slope ($\geq 0.70$) and intercept ($\leq 20$ Wm$^{-2}$) were achieved with 45 and 60 minutes averaging periods, which is consistent with literature (Gao, 2017; Leuning, 2012). The strength of linear association between ($R_n$-G) and (H+LE) around the best fit line, explained by *r* is high ($0.80 < r \leq 0.9$) at low averaging periods, i.e., 1, 5, 10 minutes, and is very high ($r > 0.9$) for other averaging periods (Table 2). However, the departure of the

data from 1:1 line is relatively low both at low and high averaging periods. Our findings show
that averaging period has minimal influence in representing the energy balance terms.
**Table 2:** Summary of linear regression parameters in closing the energy balance with different
averaging periods.

| Averaging Period | Slope | $R^2$ | Intercept (Wm$^{-2}$) | r | N | RMSE (Wm$^{-2}$) |
|---|---|---|---|---|---|---|
| 1min | 0.63 | 0.66 | 30.31 | 0.81 | 10859 | 98.38 |
| 5min | 0.59 | 0.74 | 31.56 | 0.86 | 10785 | 76.47 |
| 10min | 0.60 | 0.80 | 28.94 | 0.90 | 10753 | 64.41 |
| 15min | 0.63 | 0.84 | 26.56 | 0.92 | 7150 | 58.18 |
| 30min | 0.66 | 0.93 | 20.49 | 0.96 | 3554 | 38.33 |
| 45min | 0.70 | 0.94 | 19.99 | 0.97 | 2355 | 36.30 |
| 60min | 0.71 | 0.94 | 19.01 | 0.97 | 1765 | 35.07 |
| 120min | 0.67 | 0.93 | 20.77 | 0.96 | 811 | 39.95 |


**3.2 Effect of averaging period on EBR and EC fluxes**
The variation in energy balance ratio (EBR) with averaging period for individual
growth stages of the crop is presented in Figure 2. We observed a clear departure of EBR from





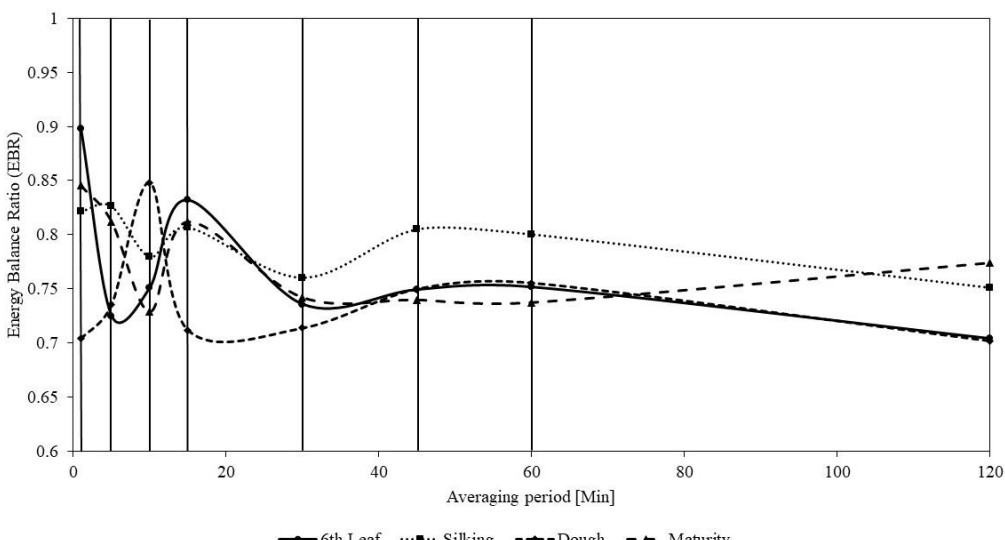


**Figure 2:** Variation in energy balance ratio (EBR) with averaging period for different growth stages. (Solid verticals from left to right correspond to the averaging periods of 1 min., 5 min., 10 min., 15 min., 30 min., 45 min., 60 min., and 120 min. respectively).

unity for all growth stages, particularly with dough and maturity stages due to ignorance of
canopy heat storage. EBR is fluctuating between 0.70 and 0.90 at low (1 − 30 min.) averaging
periods and is fairly constant (0.75 ± 0.03) at high (≥ 30 min.) averaging periods. Our reported
values of EBR during the crop growth are within the typically found range of 0.65 to 1.2 for
most of the crops (Feng, 2017; Finnigan, 2003; Wilson, 2002). The mean EBR with
conventional 30 min. averaging period is found to be 0.74, 0.76, 0.71, and 0.74 during 6[th] leaf,
silking, dough, and maturity stages respectively. Low EBR during the crop cycle can also be
attributed to the ignorance of energy transport associated with large eddies from landscape
heterogeneity, which is not captured by the EC system. Changes in daytime mean carbon and
water fluxes with averaging period for different growth stages of the crop is shown in Figure
3. Carbon fluxes (sink) have a very low mean ($1.81 \pm 0.06\ \mu\mathrm{mol\,m^{-2}s^{-1}}$) during 6[th] leaf stage,





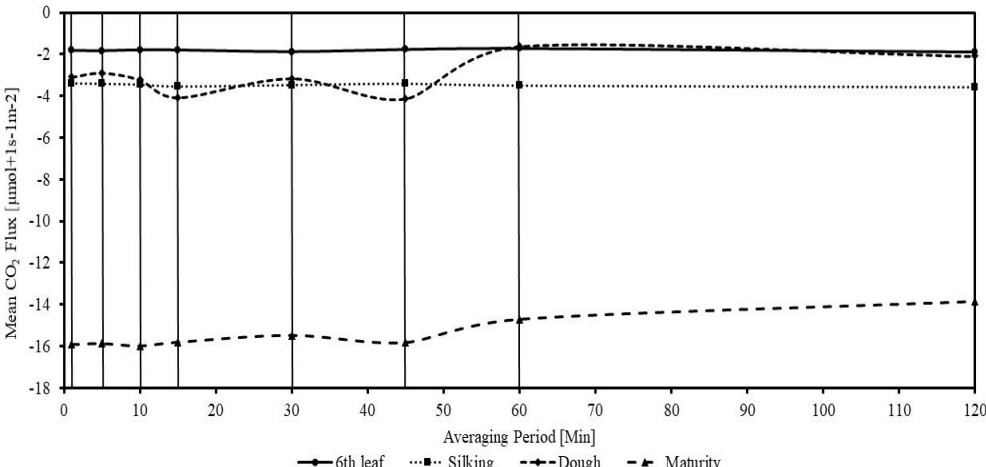


**Figure 3a:** Variation in mean carbon fluxes with averaging period for different growth stages (Solid verticals
from left to right correspond to the averaging periods of 1 min., 5 min., 10 min., 15 min., 30 min., 45 min.,
60 min., and 120 min. respectively).

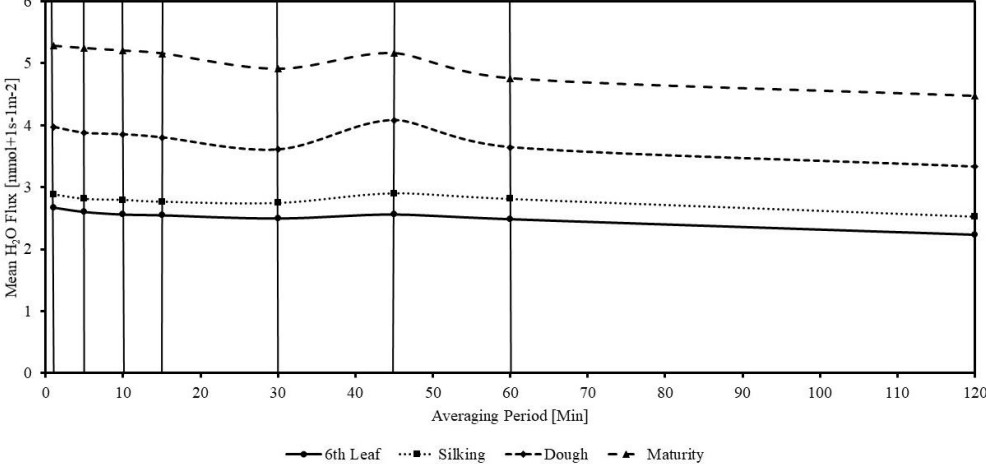

310

**Figure 3b:** Variation in mean water fluxes with averaging period for different growth stages (Solid verticals
from left to right correspond to the averaging periods of 1 min., 5 min., 10 min., 15 min., 30 min., 45 min.,
60 min., and 120 min. respectively).

low mean during silking ($3.48 \pm 0.07$ µmol m$^{-2}$s$^{-1}$) and dough ($3.03 \pm 0.87$ µmol m$^{-2}$s$^{-1}$) stages,

and a high mean ($15.44 \pm 0.75$ µmol m$^{-2}$s$^{-1}$) during maturity stage. Mean carbon fluxes during

6$^{th}$ leaf and silking stage are mostly unaffected by averaging period. We observed a gradual

increase in water vapour fluxes during the crop cycle, from 6$^{th}$ leaf ($2.52 \pm 0.13$ mmol s$^{-1}$m$^{-2}$)

to maturity ($5.02 \pm 0.29$ mmol s$^{-1}$m$^{-2}$). As the averaging period is increased, the mean water

vapour flux is decreased, with an exception at 45 min. averaging period. Distribution of error





in representing carbon and water fluxes at different averaging periods, relative to the
conventional 30 min. averaging period i.e. relative error (RE) is presented in Figure 4. During
6[th] leaf and silking stages, RE in estimating carbon fluxes is high (~ -15 %) with low averaging
periods, and is gradually diminishing towards higher averaging periods, with an exception at
very high (120 min.) average period. For dough and maturity stages, RE is found to be
significant with higher averaging periods (60-120 min). RE in estimating water vapour fluxes
is found to be insignificant at all averaging periods, irrespective of growth stage.

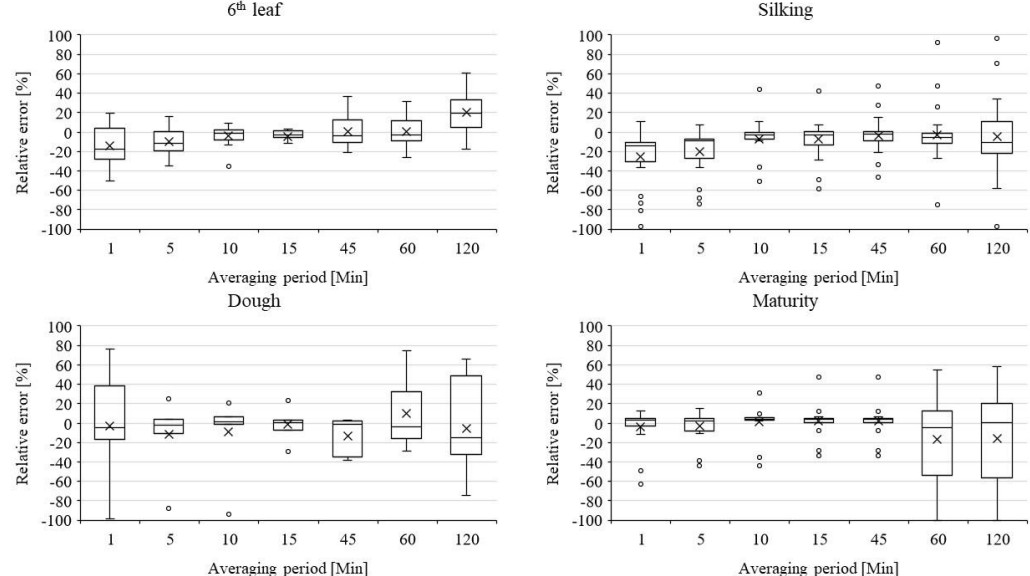

**Figure 4a:** Whisker plots showing the distribution of error in estimating carbon fluxes with various
averaging periods relative to the conventional 30 min. averaging.





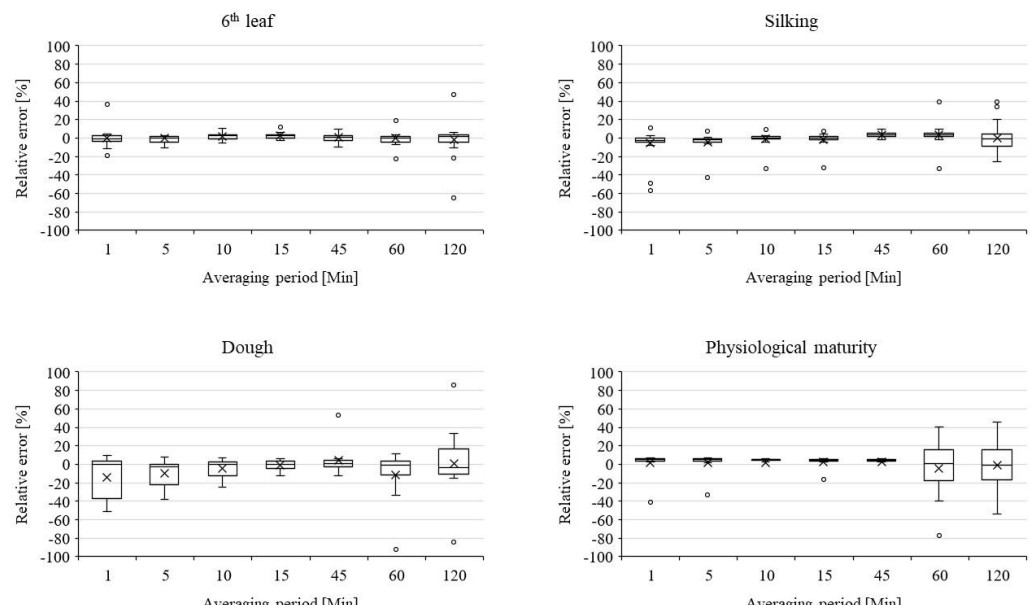


**Figure 4b:** Whisker plots showing the distribution of error in estimating water fluxes with various averaging
periods relative to the conventional 30 min. averaging.


### 3.3 Selection of Optimal averaging period

Ogive functions representing the cumulative integral of the co-spectral energy starting
with highest frequency, i.e., 0.016 Hz (T = 1 min.) for carbon and water fluxes are presented

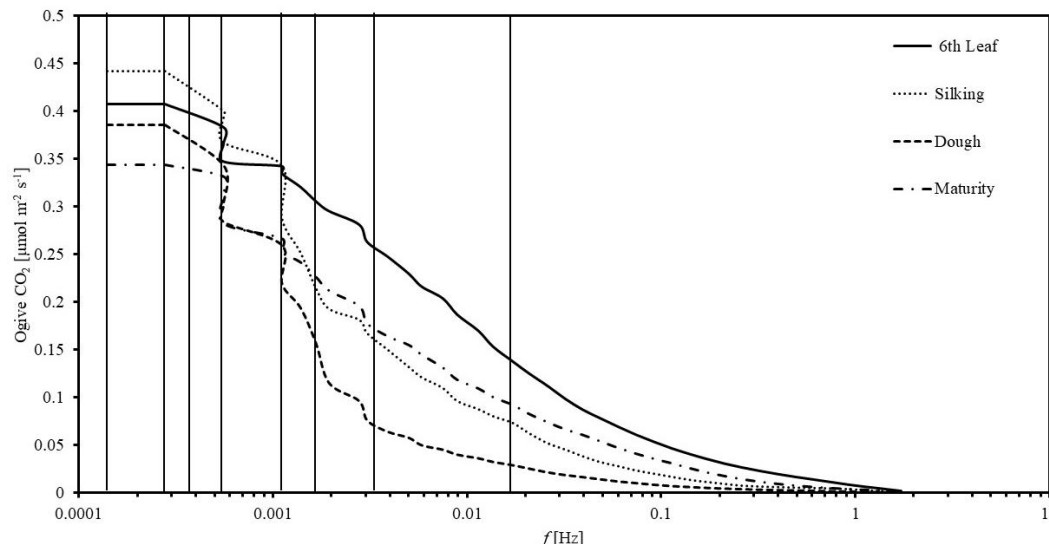






**Figure 5a:** Ogive plots of carbon fluxes for different growth stages of the Maize crop. (Solid verticals from
left to right extremes correspond to the averaging periods of 120 min., 60 min., 45 min., 30 min., 15 min.,
10 min., 5min. and 1 min. respectively).

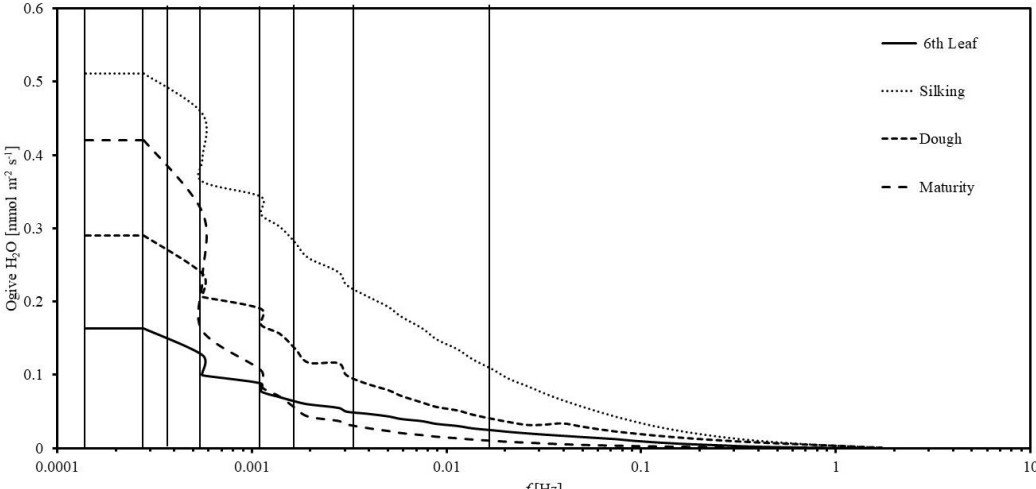


**Figure 5b:** Ogive plots of water fluxes for different growth stages of the Maize crop. (Solid verticals from
left to right extremes correspond to the averaging periods of 120 min., 60 min., 45 min., 30 min., 15 min.,
10 min., 5min. and 1 min. respectively)
in Figure 5. Shorter time periods corresponding to daytime unstable atmospheric conditions
(08:00 am to 04:00 pm) for various growth stages were investigated. Ogive plots of carbon
fluxes for $6^{th}$ leaf and silking stages showed an increasing trend upto 0.011 Hz (15 min.) and
remained fairly constant before 0.0055 Hz (30 min.). This concludes that whole turbulent
spectrum can be covered with 15 to 30 min. averaging, with negligible flux contribution from
longer frequencies. Ogive plots of carbon fluxes for dough and maturity stages showed a
continuous increasing trend without a defined plateau (horizontal asymptote) in between. This
conclude that the conventional 30 min. averaging period is inadequate to capture the low
frequency fluxes, thus demanding for higher averaging periods. We observed a similar
behaviour with water fluxes (Figure 5). The flat part of the Ogive curve representing the
optimal averaging period was found to vary across the crop cycle. While 15-30 min. time-
average is suitable for aggregating the EC fluxes during $6^{th}$ leaf and silking stages, 45-60 min.
averaging is more appropriate for dough and maturity stages. The crop biophysical factors like
LAI and plant height are minimum during $6^{th}$ leaf and silking stages contributes low quantity
of $CO_2$ and $H_2O$ fluxes (refer figure 3a & 3b) whereas they are maximum in the later stages of
the crop i.e., tasselling and maturity contributing to high quantity of $CO_2$ and $H_2O$ fluxes (refer
figure 3a & 3b). Our results are in accordance with the previous studies of Fong et al., 2020 on
Cotton, where the responses in NPP and ET were related seasonally to plant growth stages.





The previous studies on various crops revealed that the NPP and ET fluxes were initially low
in the early stages and increases towards maturity stage due to crop phenology and management
practices. To capture these low quantity fluxes, low averaging periods i.e., 15 min. is sufficient,
whereas 45 min. time-averaging period can capture high quantity fluxes that are prevalent
during later growth stages of the crop. As the crop characteristics are dependent on the crop
growth stages, a single time-averaging period is not appropriate to capture the dynamics of $CO_2$
and $H_2O$ fluxes and their ratio WUE. This clearly demonstrates that, as the plant achieves its
higher stage, flux contribution from low-frequency components becomes more valuable. Very
low averaging periods (ex: 1 min., 5 min.) were found unsuitable to capture low-frequency flux
components, which is in agreement with literature (Feng, 2017).

**3.4 Dynamics of Water use efficiency**

Daily means of water use efficiency (WUE) estimated with conventional 30 min. and

growth specific optimal averaging periods is presented in Figure 6. Mean WUE fluxes for 6th

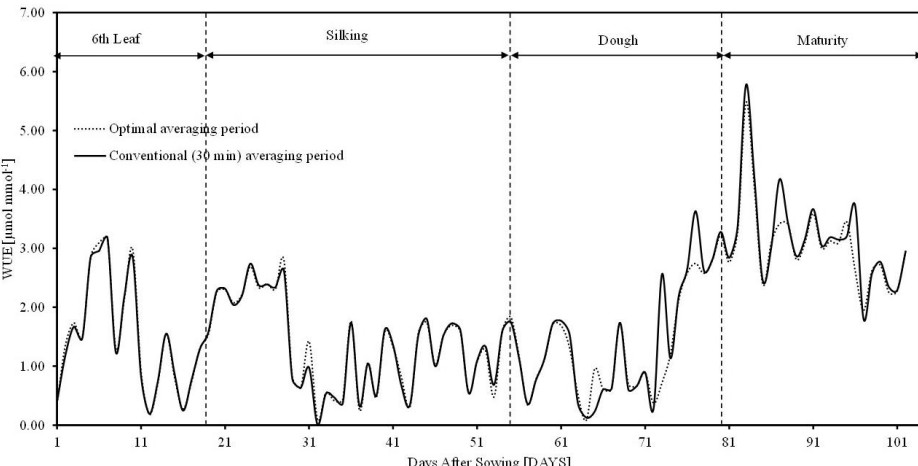


**Figure 6:** Seasonal variations in daily mean WUE fluxes obtained with conventional 30 min. (black) and
optimal averaging periods (red) during the crop cycle.
leaf, silking, dough and maturity stages with conventional 30 min. averaging are $1.48 \pm 0.96$,
$1.36 \pm 0.73$, $1.38 \pm 0.95$ and $3.184 \pm 0.78$ µmol mmol$^{-1}$ respectively. Corresponding fluxes
with stage specific optimal averaging periods are $1.49 \pm 0.95$, $1.37 \pm 0.74$, $1.39 \pm 0.79$ and $3.06$
$\pm 0.69$ µmol mmol$^{-1}$ respectively. Error in estimating mean daily WUE fluxes with 30 min.





averaging is very low (< 1.45%) during 6th leaf and silking stages, low (8.56 to 9.04 %) during
maturity stage, and is moderate (11.84 to 12.12 %) during dough stage. This conclude that,
choice of optimal averaging period is more crucial for late stage growth periods of the crop.
Distribution of error in estimating WUE fluxes with various averaging periods relative to
conventional 30 min. average period (RE) is presented in Figure 7. A close to zero RE with all
averaging periods during 6th leaf and silking stages conclude that, choice of averaging period
has insignificant role in estimating the WUE fluxes, particularly during early growth stages. A
slightly high RE (~ -5.4%) during dough and maturity stages conclude that, choice of averaging
period matters for WUE estimation during late stage periods. Hence, conventional 30 min.
averaging period can be considered for estimating WUE fluxes during 6th leaf and silking
stages, whereas optimal averaging period need to be considered for estimating WUE fluxes
during dough and maturity stages. Correlation charts showing the linear association within
carbon, water, and WUE fluxes represented at different averaging periods is presented in Figure
8. For ease with comparison, data for the entire crop cycle was considered. Linear association

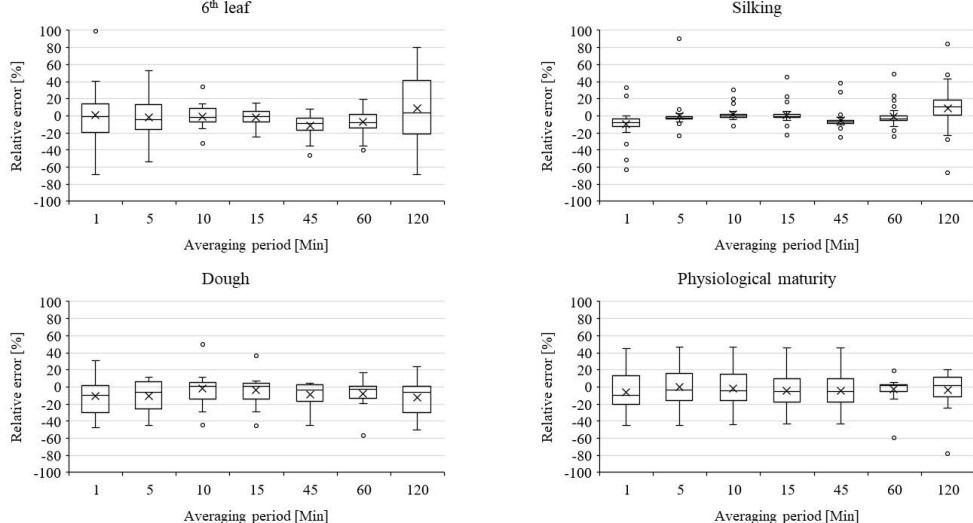


**Figure 7:** Whisker plots showing the distribution of error in estimating WUE fluxes with various averaging

periods relative to the conventional 30 min. averaging.
between any two averaging periods is positive ($\rho > 0.56$) for carbon and water fluxes. Except
with 120 min. time-averaging, all other averaging periods are strongly correlated ($\rho > 0.87$)
with 30 min. averaging period. However, a poor linear association in WUE fluxes was observed
between any two averaging periods. This conclude that, the need for optimal averaging period
is more crucial in estimating WUE fluxes rather than individual carbon and water fluxes. Our





findings can improve representation of WUE fluxes using EC data, thereby help in developing
efficient water management strategies in response to WUE changes.

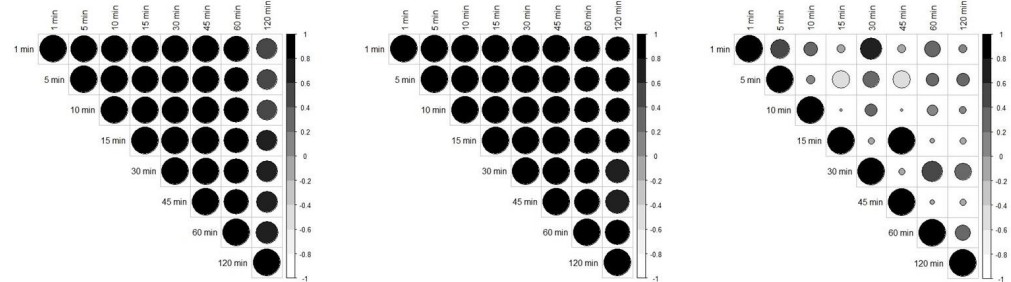


**Figure 8:** Correlation charts showing the linear association of **a**) Carbon fluxes, **b**) Water fluxes, and **c**)
WUE fluxes estimated with different averaging periods.

**4.0 CONCLUSIONS**
This study explores the effect of averaging period of EC fluxes on EBR dynamics and
WUE in semi-arid Indian conditions. The proposed methodology was applied on drip-irrigated
maize field for one crop period (May-Sept 2019). Major findings of this study are:
- EBR was found vary marginally at low averaging periods and less significant during
higher averaging periods.
- With reference to conventional 30 min. averaging period, relative error is within 12%
for 10-45 min. averaging periods for carbon fluxes and is within 5% for 15-45 averaging
periods for water fluxes.
- From ogive analysis we found the optimal averaging period as 15 - 30 min. for the 6th
leaf, and silking stages, and as 45 – 60 min. for the dough and maturity stages.
- The mean carbon fluxes are increasing from $1.81 \pm 0.06$ $\mu mol^{+1}m^{-2}s^{-1}$ (6th leaf stage)
to $15.44 \pm 0.75$ $\mu mol^{+1}m^{-2}s^{-1}$ (maturity stage) which indicates that carbon sink is a
function of crop growth period. In case of water fluxes, it increased from $2.52 \pm 0.13$
$mmol^{+1}m^{-2}s^{-1}$ (6th leaf stage) to $5.02 \pm 0.29$ $mmol^{+1}m^{-2}s^{-1}$ (maturity stage). Variation of
carbon and water fluxes are directly influencing WUE dynamics.
- The variation in WUE was increased subsequently with the plant growth and achieved
its maximum value of 5.17 $\mu mol$ $mmol^{-1}$ in between dough to maturity stages which
concludes that, crop consumes more carbon than water as the crop period progresses.





- The correlation between $CO_2$ and $H_2O$ fluxes for all averaging periods was found to be high. However, WUE, which is calculated as the ratio of $CO_2$ and $H_2O$ fluxes, is not following the same pattern. While 45 min. and 15 min. averaged WUE exhibits a good correlation, 30 min. averaged WUE is not correlated with other averaging periods. Averaging period is found to be an influencing factor in controlling WUE, hence should be considered with caution during the crop growth.

This study is limited to understand the role of different time-averaging periods on EC observed carbon, water fluxes as well as EC derived WUE fluxes. Study findings can help to accurately characterise WUE of Maize crop considering growth stages for effective implementation of irrigation strategies.

**Data Availability Statement:**

All footprint climatologies, site-level data files, and supplementary material can be accessed via the Zenodo Data Repository (https://zenodo.org/badge/latestdoi/528291820) (Shweta07081992, 2022)

**Author Contribution:**

**Arun Rao Karimindla**: Data processing and Analysis, Writing- Original draft. **Shweta Kumari:** Conceptualization, Methodology, Project Supervision. **Saipriya SR**: Data processing Analysis, and Writing- Original draft. **Syam Chintala:** Data processing and Analysis, Writing- Original draft. **BVN Phanindra Kambhammettu:** Project Administration, Writing- Reviewing and Editing.

**Competing interests:**

The authors declare that they have no known competing interests or personal relationships that could have appeared to influence the work reported in this paper.

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
