# Peer review of "dynamics of Maize crop"

_Atmospheric Measurement Techniques, 2023_

## Author Comment (AC1)

Manuscript Number: AMT-2023-253

Manuscript Title: Role of time-averaging of eddy-covariance fluxes on water use efficiency dynamics of Maize crop.

**AUTHORS' RESPONSES TO REVIEWER – 2 COMMENTS**

**General comments**

The authors present a study of the impact of eddy covariance (EC) averaging time on estimation of water use efficiency (WUE). While the impact of averaging of eddy covariance flux results has been extensively studied, the impact on WUE specifically has not. Therefore, the manuscript provides a contribution to broadening understanding of the important aspect of EC flux processing on results.

- Thanks to the reviewer for highlighting the importance and need of the given study.
- We have done additional analysis in arriving at the optimal averaging period using Ogive plot for WUE fluxes, and the results are provided in the revised manuscript (lines: 394 to 401, Section 3.3 and Figure 5c).

The authors perform analysis of how different averaging times (varying over a broad range of 1, 5, 10, 15, 30, 45, 60, and 120 minutes) affect the results during different stages of Maize crop development. The main finding of the study is that, whereas the commonly applied 30 min averaging is a good choice for most of the conditions, the longer averaging times yield better results during the dough development stage. The need for longer averaging period must result from different prevailing observational and/or meteorological conditions. The authors have not analyzed the underlying main drivers that determine the need for longer averaging time. They have suggested that canopy heterogeneity might be one of the reasons. I suggestion to perform additional analysis of prevailing conditions (minimum wind speed and direction, which could hind also impact of heterogeneity, and stability) during different canopy stages to be able to make link with optimal averaging times.

- Thanks to the reviewer for an insightful thought.
- Following reviewer suggestion, temporal trends in 'wind speed' and 'wind direction' were plotted for different time-averages (1 to 120 min), and the results are presented below.

[Figure]

*Figure 1: Temporal variations in wind speed (U) and wind direction during the crop period, considering different averaging periods (1 to 120 min)*

- Transport of water and carbon fluxes carried by vertical wind speed (eddies) is highly fluctuating between vegetation and atmosphere. Hence, time-averaging of these fluxes (water and carbon) have resulted in different profiles (Figures: 3a and 3b).

- However, time-averaging has no impact on 'wind speed' and 'wind direction' profiles (Figure 1 above) as these are relatively stationary over the time-periods considered. This point was mentioned in the revised manuscript (line: 325).

  "We could not observe any significant differences in temporal trends of 'wind speed' and 'wind direction' between averaging periods, hence meteorological conditions were not analysed by varying the time-average"

- However, we observed that, optimal averaging period is inversely related wind speed variation.

- For example, $6^{th}$ leaf and silking stages, where the variation in wind speed is high ($2.04 \pm 0.55$ m s$^{-1}$; $3.66 \pm 0.96$ m s$^{-1}$) have resulted in a shorter time averages (15 min). Similarly, Dough and maturity stages, where the variation in wind speed is low ($3.54 \pm 0.57$ m s$^{-1}$; $3.19 \pm 0.42$ m s$^{-1}$) have resulted in a longer time averages (45 min).

- We could not observe the role of wind direction in selecting the optimal time average, as wind direction is found to be constant ($100 \pm 0.5^{0}$) throughout the crop period, except for the $6^{th}$ leaf stage.

One important clarification is needed regarding the detrending and averaging. Section 2.2: Did you use linear detrending and then block averaging? Since linear detrending performs as additional high-pass filter then this is very important to be very specific and emphasize also in Abstract and Conclusions. Without linear detrending the optimal averaging times could be different.

- Sorry for the confusion created.
- Detrending was performed to obtain the turbulent fluxes (by subtracting mean from the instantaneous values).
- We considered either 'block averaging' or 'linear trend removal' for detrending (Burba, 2022), but not both.
- This sentence is re-phrased in the revision document (lines: 168 to 169) as below:
  "Either block average method or linear trending method were considered to compute the turbulent fluctuations"
- Appropriate detrending method was used for carbon and water flux computation, and this is mentioned in lines: 169 to 173 of the revised manuscript as below:
- "Block averaging method was used for detrending the fluxes at 1, 5, 10, 30, 45, and 60 min averaging periods. Longer averaging periods (e.g. 120 min) has resulted in inconsistency in the obtained fluxes, which is a weakness of the block averaging (Renhua, 2005; Sun et al., 2006). Hence, linear trend removal method was used to compute the fluxes for 120 min averaging period".

The main emphasis of the manuscript is to evaluate the impact of the averaging time on WUE. Please also conclude if the choice of averaging time for accurate determination of WUE is different from energy and carbon fluxes (fluxes of scalars).

- Thanks to the reviewer for this valuable suggestion.
- Following reviewer suggestion, we estimated WUE fluxes for different time-averages (1 min to 120 min), and correspondingly plotted 'Ogive plots' of WUE considering different averaging periods (Figure 5c of revised manuscript).
- Interestingly, we arrived at the same optimal averaging periods (15 min for 6[th] leaf and Silking stages; 45 min for dough and maturity stages), as we observed for carbon and water fluxes.

- We have added the above figure (Figure 5c) and related text in the results section (lines: 394 to 401) of the revised manuscript.

**Detailed comments**

1. 9. The low-frequency flux inclusion is not the only factor and not under all observation conditions that might affect the accuracy of the EC flux estimates. Please be more specific with statement.

- Error sources that affect accuracy of EC fluxes are grouped into:
    1) Unrepresentative (due to footprint heterogeneity, unsatisfied underlying theory)
    2) Measurement uncertainties (due to random errors, interference and contamination, sensor drifts)
    3) Measurement biases in fluxes (tilt, frequency losses, air density fluctuations etc)

- Among these, we considered the effect of "frequency losses" alone in this study. This is also because, a majority of error sources are either unavoidable or uncontrollable.

- This was mentioned in the Introduction section of the revised manuscript (lines: 68 to 72) as below:

"Error sources that affect the accuracy of EC fluxes are grouped into: i) Unrepresentative (due to footprint heterogeneity, unsatisfied underlying theory), ii) Measurement uncertainties (due to random errors, interference and contamination, sensor drifts) and iii) Measurement biases in fluxes (tilt, frequency losses, air density fluctuations etc)."

- Since Abstract is the concise version of the entire work, only the applicable cause is mentioned.

2. 13-14. Canopy heat storage should net be a significant factor over a relatively long period of time.

- We politely disagree with the reviewer.

- A high canopy cover (LAI > 3) was observed during the dough and maturity stages (Table 1). Ignorance of this canopy storage term ($\Delta S$) is one significant cause of low energy balance closure (EBC).

- Since energy balance is calculated on a daily basis, canopy storage is a significant sink, resulting in lower EBR (Figure 2).

3. 16-18: what were the main driving factors that the optimal averaging time differed for different stages of canopy development? See my main comment.

- Please refer to our detailed response against main comment.

- Referring to Figure 1 above, it can be concluded that, the choice of optimal averaging period is related to wind speed. For example, 6th leaf and silking stages, where the variation in wind speed is high ($2.04 \pm 0.55$ m s$^{-1}$; $3.66 \pm 0.96$ m s$^{-1}$) have resulted in a shorter time averages (15 min). Similarly, Dough and maturity stages, where the variation in wind speed is low ($3.54 \pm 0.57$ m s$^{-1}$; $3.19 \pm 0.42$ m s$^{-1}$) have resulted in a longer time averages (45 min).

- We could not observe the role of wind direction in selecting the optimal time average, as wind direction is found to be constant ($100 \pm 0.5^0$) throughout the crop period, except for the 6th leaf stage.

- However, it can be observed that time-averaging has no affect on the temporal trends of 'wind speed' and 'wind direction (Figure 1 above). For this reason, wind speed and wind direction were not presented / analysed for different averaging periods.

4. 32. The abstract states the error compared to 30 min averaging was marginal except for dough stage. Be more specific, e.g. 30 min averaging is not sufficient for all conditions.

- Agree with the reviewer, that 30 min averaging is not sufficient for all conditions.

- However, 30-min is the widely accepted conventional time-averaging period in EC flux estimation.

- To highlight the importance of using optimal averaging period, we analysed the error in representing the fluxes.

- To be more specific, we have added the following sentence in the abstract (lines: 24 to 26) "Error in representing WUE with conventional 30 min averaging is marginal (< 1.5 %) throughout the crop period except for the dough stage (12.12 %). We conclude that the conventional 30 min averaging of EC fluxes is not appropriate for the entire growth stage".

5. 34. The sentence is missing some word, for example "Different averaging time need to be used following the crop growth stage".

- Agree with the reviewer, the given sentence is slightly confusing.

- Research highlight 4 is now modified for a better readability, as follows:

"Different time averaging periods are to be considered to compute the EC fluxes considering the crop growth stage".

6. 51, the symbol colon (:) looks redundant after "water productivity"

- Corrected (line: 103 of revision manuscript).

7. 61-62, readability would benefit from re-arranging the parenthesis, e.g. "WUE is estimated as the ratio of gross primary product (GPP: proxy for photosynthesis) to evapotranspiration (ET: proxy for water consumption).

- Agreed. The sentence is modified as:
  "WUE is estimated as the ratio of net primary product (NPP: proxy for photosynthesis) to evapotranspiration (ET: proxy for water consumption)".

8. 112, the average +- error after "Temperatures are high during summer" and "low during winter": what do these errors represent?

- No, these are not the errors.

- We presented the data in the form of ($\mu \pm \sigma$), where $\mu$ denotes data mean, and $\sigma$ denotes one-standard deviation.

- It is convention in statistics to present the data variability in ($\mu \pm \sigma$) format, that provides average amount of variability in the datasets.

9. 142-143: Did you use linear detrending and then block averaging? Also, I assume this was "to derive" turbulent fluctuations and not "to correct".

- Sorry for the confusion.

- We considered either 'block averaging' or 'linear trend removal' for detrending (Burba, 2022).

- This sentence is re-phrased in the revision document (line: 168) as below:
  "Either block averaging method or linear trending method were considered to compute the turbulent fluctuations"

- Appropriate detrending method used for carbon and water flux computation is mentioned in lines: 168 to 173 of the revised manuscript as below:

- "Block averaging method was used for detrending the fluxes at 1, 5, 10, 30, 45, and 60 min averaging periods. Longer averaging periods (e.g. 120 min) has resulted in inconsistency

in the obtained fluxes, which is a weakness of the block averaging (Renhua, 2005; Sun et al., 2006). Hence, linear trend removal method was used to compute the fluxes for 120 min averaging period".

- In order to preserve the low frequency flux loss during averaging, we applied linear trend removal method for 120 min averaging.

10. 151, what is friction velocity correction? Do you mean filtering of night-time observations according to friction velocity threshold? Be specific here.

- Yes, we applied friction velocity (u*) correction to filter out the night time observations by specifying a velocity threshold ($> 0.25$ m s$^{-1}$).

- This was mentioned in the revision (line: 179) as follows:

"There is a need to perform secondary corrections on the data that include flux spike removal (Vickers & Mahrt, 1997), friction velocity corrections (to filter night time observations), gap filling and uncertainty analysis (Finkelstein et al. 2001), skewness & kurtosis removal, spectral corrections, and frequency corrections."

11. 159: lack of conservation should be "lack of energy balance closure".

- Sorry for any confusion. We modified this sentence (line: 189) as follows:

"Violation of law of conservation of energy resulting from the EC observed energy terms is referred as energy balance closure (EBC)"

12. 168: where this specific threshold EBC $>= 0.7$ comes from that ensures reliability of EC fluxes? Please be more specific and/or provide references.

- The threshold for EBC ($> 0.7$) is used to comment on the reliability of EC fluxes. A number of studies such as Barr et al., 2006 and Kidston et al., 2010 have considered a similar threshold ($0.7 \pm 0.03$) under unstable day time periods.

- As suggested, references were provided in the revision (line: 206)

"A high EBR (EBR $\geq 0.7$) ensures reliability of EC observations for use with flux estimation (Barr et al., 2006; Kidston et al., 2010)."

13. 180: the main challenge with real-world data is data the spectral gap is obscure or difficult to identify. Otherwise, the choice of the averaging time would be simple task.

- Agree with the reviewer.

- For this reason, we considered Ogive method to choose the optimal averaging period to compute the fluxes.

14. 195, also section 2.2, did you perform coordinate rotation at the same time interval basis as the averaging?

- Yes, we performed double coordinate rotation at the same interval as averaging period.
- This was clarified in the revision (lines: 167 to 168) as follows:

"Tilt corrections were made by the double axis rotation method for each averaging period".

15. 203-204: how did you define the optimal averaging period?

- For ease with understanding, we added the following point in arriving at optimal averaging period using the Ogive plot (lines: 241 to 242)

"In other words, the point at which the Ogive plot flattens out represents the optimal averaging period"

16. 207, eq. (5), since this is RMSE error, the square root should be taken from the value in squared? Which is missing in the expression.

- Sorry, this is typo. We modified the expression as:

$$RMSE = \left[ \frac{\sum_{i=1}^{n}((R_n - G)_i - (H + LE)_i)^2}{n} \right]^{0.5}$$

17. 221-222, fig 1. Denote the subplots with relevant averaging times. Currently it is not possible to follow which plot corresponds to what averaging time.

- Thanks for pointing this.
- Each sub-plot of Figure 1 now contains the respective averaging period, in the inset.

18. 267, and the main comment: please analyse the potential impact of meteorological conditions (wind speed and direction, stability). The wind direction variability might provide better insight related to landscape heterogeneity; currently this remans a hypothesis.

- Thanks for the important suggestion.
- This point was clarified in the response under main comment.

- We performed additional analysis on the role of meteorological conditions (wind speed, wind direction) on $CO_2$, $H_2O$ and WUE obtained using optimal time average periods. Here are the key observations / findings:

-  From the dynamics of wind speed, it is observed that the mean wind speed was initially low ($2.04 \pm 0.55$ m s$^{-1}$) in 6$^{th}$ leaf stage and then starts increasing and reached a mean of $3.66 \pm 0.96$ m s$^{-1}$s in silking, $3.54 \pm 0.57$ m s$^{-1}$ in dough stages after that it slowly decreasing towards the end of crop stages and finally reached a mean of $3.19 \pm 0.42$ m s$^{-1}$ in the maturity stage which is obtained using optimal time averages i.e. 15 min and 45 min.

- We observed high fluctuations of wind speed in the initial stages i.e. 6$^{th}$ leaf, silking stages and these fluctuations were slowly stabilized at end of the crop stage, towards winter months. The optimal time average of 15 min is able to capture the random fluctuations in the wind speed.

- We could not observe the role of wind direction in selecting the optimal time average, as wind direction is found to be constant ($100 \pm 0.5^{0}$) throughout the crop period, except for the 6$^{th}$ leaf stage.

- Regarding stability, we only considered day-time unstable atmospheric conditions (08:00 am to 04:00 pm) which are the active photosynthetic hours for carbon uptake.

- We have not considered the effect of landscape heterogeneity (if any). Our hypothesis is: During unstable atmospheric conditions, flux footprint is relatively smaller, hence completely contributed by the homogenous maize crop. This is one limitation of our research, which is mentioned in the revision document as follows:(lines: 485 to 489)

"This study is limited to understand the role of different time-averaging periods on EC observed carbon, water fluxes as well as EC derived WUE fluxes contributed by homogeneous Maize crop which is having relatively smaller flux footprint in an unstable atmospheric condition".

19. L. 361, Fig. 8: what does the circle size represent? Also, how did you compare e.g. 45 min and 30 min averaging (45 min period does not fully overlap with 30 min period)? Plot c) looks inconsistent (or difficult to interpret). How do you interpret that for carbon dioxide and water the correlations between different averaging times are all very good but for WUE not. One would expect that closure averaging times (for example 45 min and 30 min) correlate better than more different (e.g. 45 min and 15 min). Could the specific "pattern" of this plot be the result of periods mismatch?

- The circle size represents the value (strength) of "r" in proportion to the size of the square box.

- A larger size of the circle between any two averaging periods denotes a high correlation strength between the two datasets, and vice-versa.

- Rather than inconsistent, this is an interesting finding. Though carbon and water fluxes are strongly correlated individually, their ratio term (WUE fluxes) is poorly correlated between any two averaging periods.

- For any two averaging periods (refer link: https://zenodo.org/badge/latestdoi/528291820), variation in carbon sink and water vapour are low, whereas variation in WUE is high.

- For example, the following table provides the carbon, water, and WUE fluxes at 15 min and 45 min averaging periods for some selective dates.

- Observe the huge differences in WUE (5.42 to 20.21 %) in comparison to the differences in carbon (0.14 to 10.37 %) and water (0.25 to 8.60 %) fluxes.

- Hence, variation in WUE between any two time-averages is much larger than variation in carbon or water fluxes.

- For this reason, we are highlighting that, choice of optimal time-averaging is crucial in WUE analysis, rather than carbon or water flux analysis.

| Day | $CO_2$ [$\mu mol^{+1}\ s^{-1}\ m^{-2}$] | | Deviation in $CO_2$ Flux [%] | $H_2O$ [$\mu mol^{+1}\ s^{-1}\ m^{-2}$] | | Deviation in $H_2O$ Flux [%] | WUE [$\mu mol\ mmol^{-1}$] | | Deviation in WUE Flux [%] |
|---|---|---|---|---|---|---|---|---|---|
| | 15 min | 45 min | | 15 min | 45 min | | 15 min | 45 min | |
| 1 | -0.3719 | -0.37144 | 0.1415 | 1.4518 | 1.4555 | 0.2571 | 0.3714 | 0.29669 | 20.1315 |
| 2 | -0.7048 | -0.63173 | 10.3755 | 1.1326 | 1.1164 | 1.4354 | 1.3640 | 1.08831 | 20.2168 |
| 3 | -1.4137 | -1.4207 | -0.4905 | 1.0586 | 1.1497 | 8.6037 | 1.7367 | 1.48193 | 14.6742 |
| 4 | -0.7963 | -0.7929 | 0.4221 | 0.9210 | 0.9152 | 0.6357 | 1.4913 | 1.41033 | 5.4297 |

- Also note that, the correlation strengths were plotted considering the entire crop cycle dataset, rather than individual growth stages.

- These causes are explained in the revised manuscript (lines: 448 to 451) as follows:

"However, a poor linear association in WUE fluxes was observed between any two averaging periods, which is attributed to a larger variation in individual WUE fluxes between averaging

periods. However, the corresponding individual carbon and water fluxes have recorded low variations between time averages".

---

## Author Comment (AC2)

Manuscript Number: AMT-2023-253

Manuscript Title: Role of time-averaging of eddy-covariance fluxes on water use efficiency dynamics of Maize crop.
* * *
**AUTHORS' RESPONSES TO REREREE – 3 COMMENTS**
* * *
The current study utilizes micrometeorological measurements to evaluate the effect of the averaging time on turbulent flux estimates obtained by the Eddy Covariance (EC) method. This topic has been extensively explored over the last few decades. However, such investigations on sites with different characteristics are always relevant. The manuscript novelty is the determination of optimal averaging time in a drip-irrigated maize field site. The remarkable result is the determination of the optimal averaging time for the different plant growth stages. The turbulent fluxes, estimated using the optimal values, are compared with those obtained using the usual value of 30 min, widely used in unstable daytime periods. The authors also showed how the improvement in flux estimates affects the values of related variables such as water use efficiency (WUE).

The current version of the manuscript is well structured. All sections are also clearly presented. However, I suggest some questions and corrections that should be addressed to improve the overall manuscript.

- Thanks to the reviewer for a comprehensive review by highlighting the novelty and efforts in shaping the manuscript.
- All suggestions / comments were duly addressed in the revised manuscript.

**I - Geral comments:**

1) As the author mentioned in the manuscript, "Optimal averaging period (T1) should be long enough to reduce random error (Berger, 2001) and short enough to avoid non-stationarity associated with advection (Foken & Wichura, 1996)". In summary, the core of the presented analysis is how accurate the eddy covariance method is to account for the transport associated with eddies of different sizes. Generally, the eddies sizes in an unstable boundary layer strongly depend on the wind stress and sensible heat flux at the vegetated surface, which in turn depend on the overall characteristics of the rough elements in the surface (plants). This is briefly mentioned in the text (lines 229–231) but is not linked with the obtained results.

- We partially agree with the reviewer. The core of the analysis is to identify 'optimal time-average' to estimate carbon and water (hence, WUE) fluxes.
- Also note that, the physiological factors are not observed at the same scale as that of the flux observations, hence we could not perform any statistical analysis.
- Since these factors (such as plant height, LAI) vary across the growth-stages, we have considered growth-stage specific optimal time averages for EC flux analysis.
- This point is already presented in the manuscript (lines: 245 to 249)

"The biophysical and physiological characteristics such as plant height, crop water requirement, LAI, etc. changes with respect to the crop growth stage (Chintala et al., 2024) and have a significant effect on the EC fluxes. Since these factors vary over growth stages, time-averaging of EC fluxes is separated based on crop growth stage."

2) Turbulent transport in an unstable (daytime) boundary layer is dominated by large convective eddies with time scales larger than 15-20 minutes [1]. Is there a reason to include short averaging times (1, 5, 10, and 15 min) in the analysis?

- The objective of this study is to understand the role of time-averaging on carbon, water fluxes (hence, on WUE fluxes) during the Maize crop growing period, and also to arrive at the optimal averaging period for use with analysis.
- This is clearly mentioned in the abstract (line: 11) and introduction (line: 119) sections of the manuscript.
- For this reason, we considered both short-term (1, 5, 10, 15 min), conventional (30 min), and long-term (45, 60, 120 min) averaging periods for flux estimation. Ogive optimization was then used to fine-tune the periods of flux calculations.
- Shorter average periods can help in preserving the stationary of data series (Sun et al., 2007), and effectively represent turbulent spikes.

**II - Specific comments:**

1) Lines 139-140: IRGASON is the model of the integrated system anemoter 3D – IRGA. As mentioned in the text, it sounds like two sets of instruments are used.

- Sorry for the confusion. We used the integrated $CO_2$-$H_2O$ open-path gas analyzer and 3D Sonic anemometer, with model number "IRGASON-EB-NC"

- We modified the given sentence as: "The flux system is composed of integrated $CO_2/H_2O$ open-path gas analyzer and 3D sonic anemometer (IRGASON-EB-NC, Campbell Sci. Inc., USA) to measure $CO_2$ and $H_2O$ concentrations at 3 m above the canopy".

- This is reflected in lines: 143 to 146 of revised manuscript.

2) Line 140: "to measure CO2 and H2O fluxes …". These instruments do not measure fluxes directly but wind speed components, air temperature, and H2O and CO2 concentrations.

- Agree, and we modified the sentence as: "The flux system is composed of integrated $CO_2/H_2O$ open-path gas analyzer and 3D sonic anemometer (IRGASON-EB-NC, Campbell Sci. Inc., USA) to measure $CO_2$ and $H_2O$ concentrations at 3 m above the canopy".

- This is reflected in line: 143 to 146 of revised manuscript.

3) Line 163: "The block average method and linear detrending method". The authors used a series of corrections and quality control tests to ensure the best flux estimates. However, the block average acts as a high-pass filter depending on the block size, neglecting the fluxes associated with the low-frequency fluctuations. This influences the results obtained in the paper. What is the block size used for this analysis?

- We considered either 'block averaging' or 'linear trend removal' for detrending (Burba, 2022), to compute the turbulent fluxes.

- While 'block averaging' is used for 1, 5, 10, 15, 30, 45, and 60 min averaging periods, linear trending method was used for 120 min averaging period.

- Please note that: block averaging smoothens the variations of signal within each block, where as a high-pass filter removes low-frequency components from the signal. Hence, the two are not the same (Massman, 2000; Peltola et al., 2021).

- The block size used for analysis is equal to averaging period.

- Since we observed inconsistency in the obtained fluxes at higher time averages (120 min), we applied linear detrending method for 120 min time averaging period.

- The same was mentioned in the modified manuscript (lines: 169 to 173) as below:

- "Block averaging method was used for detrending the fluxes at 1, 5, 10, 30, 45, and 60 min averaging periods. Longer averaging periods (e.g. 120 min) has resulted in inconsistency in the obtained fluxes, which is a weakness of the block averaging (Renhua, 2005; Sun et

al., 2006). Hence, linear trend removal method was used to compute the fluxes for 120 min averaging period".

4) Line 194: equation (2) – This equation defines the mass fluxes, as represented in eqs. (3) and (4), used to obtain WUE. To avoid being misunderstood, it is also useful to define the expressions for the turbulent energy fluxes (Le and H) used in the definition of EBR (eq. (1)).

- Agree with the reviewer, and the expression for turbulent fluxes is added in the revised manuscript while discussing the terms of EBR (lines: 199 to 200)
- where, sensible heat is computed as:

  $H = \rho_a \, C_p \overline{w'T'}$

  and latent heat is computed as:
- $LE = L_v \overline{w'\rho_v'}$ (energy form, J m$^{-2}$ s$^{-1}$ or W m$^{-2}$)
- where $\rho_a$ is the air density; $C_p$ is the specific heat of air, $w'$ is the wind velocity fluctuation, $T'$ is the temperature fluctuation, $L_v$ is the latent heat of vaporization and $\rho_v'$ is the H$_2$O gas concentration fluctuation.

5) Line 243: equation (7) The therm [(Rn-G)i – (H+LE)i] should be squared to ensure real values by the root squared.

- Sorry, this is typo. We modified the expression as:

  $$RMSE = \left[ \frac{\sum_{i=1}^{n}((R_n-G)_i-(H+LE)_i)^2}{n} \right]^{0.5}$$

6) Figure 1: To increase figure quality. The axis labels and ticks are too small. The author should indicate which subplot corresponds to each averaging time.

- Thanks for pointing this.
- Each sub-plot of Figure 1 now contains the respective averaging period, in the inset.

7) Line 281: What does "r" represent? It was not defined before.

- 'r' represents the Pearson correlation coefficient
- Since 'r' is a widely used statistical metric, it was not defined earlier.
- However, for the benefit of the reader, expression for 'r' is given in section 2.4 of modified version.

$$r = \left\{ \frac{\sum_{i=1}^{n}\left[(R_n-G)_i - \overline{(R_n-G)}\right]\left[(H+LE)_i - \overline{(H+LE)}\right]}{\sqrt{\sum\left[(R_n-G)_i - \overline{(R_n-G)}\right]^2\left[(H+LE)_i - \overline{(H+LE)}\right]^2}} \right\}$$

8) Line 283: I suggest using "short" and "long" averaging periods.

- We used 'short' and 'long' averaging periods consistently throughout the manuscript.

9) Line 283: "Our findings show that averaging period has minimal influence in representing the energy balance terms". This sentence is true on average for several days (and different plant stages). Individually, the averaging time effects the components of the energy balance, as represented by the large scatter in the inset plots for the short average times.

- Sorry for any confusion.
- A high correlation ($r > 0.8$) is observed between available energy ($R_n$-G) and turbulent fluxes (H+LE) for all averaging periods. This conclude that, averaging period has minimal influence in representing the energy balance terms.
- However, a high data scatter around 1:1 line for shorter averages (Figure 1) is due to: large sample size, and data randomness.
- This is clarified in the revision (lines: 302 to 305) as below:

"Our findings show that averaging period has minimal influence in representing the energy balance terms. However, data scatter around 1:1 line is high for shorter time-averages due to large sample size and data randomness".

10) Line 301: "Low EBR during the crop cycle can also be attributed to the ignorance of energy transport associated with large eddies from landscape heterogeneity, which is not captured by the EC system". Can you explain this hypothesis better? Is there such landscape heterogeneity at the studied site? In this analysis, the mentioned large addies should have characteristics time scales greater than 60–120 minutes to not be taken into account by the EC method. Is there another reason why the EC system fails to account for such eddies?

- Land scape heterogeneity we mean refers to the 'spatial composition of land-parcels within the footprint of the EC flux tower'
- EC method assumes the landscape within the footprint of measurement is flat and homogenous.
- Any violation results in landscape heterogeneity, thus affecting the EC fluxes and EBR.

- We have not considered the effect of landscape heterogeneity (if any). Our hypothesis is: During unstable atmospheric conditions, flux footprint is relatively smaller, hence completely contributed by the homogenous maize crop. This is one limitation of our research, which is mentioned in the revision document (lines: 322 to 325) as:
  "Low EBR during the crop cycle can also be attributed to the ignorance of energy transport associated with large eddies from landscape heterogeneity. However, EC method assumes the landscape within the footprint of measurement to be flat and homogenous. This violation might have lowered the EBR."

11) Line 325: "RE in estimating water vapour fluxes is found to be insignificant at all averaging periods, irrespective of growth stage." Again, it is important to emphasize that this is true for the averaged result (several days), not for each individual flux measurement. Figure 4b) (dough and maturity) shows the large variability in the relative error determined using either shorter or longer averaging times.

- Agree with the reviewer, we have provided a general / concise finding, which is not appropriate for all time-averaging periods.
- The consideration of averaged result is provided in lines: 347 to 348 of the revision as:
- "The RE is obtained by considering daily averages in the deviations for each growth stage."
- Variability in RE for water vapour fluxes is provided in lines: 351 to 354 of the revision as:
- "RE in estimating water vapour fluxes is found to be insignificant at all averaging periods for the 6th leaf and silking stages. However, dough and maturity stages have shown a large variation in RE considering either too-short (1, 5 min) or too-long (60, 120 min) time averages".

12) Figures 4a) and 4b) (dough and mature stages): Following the above argument, the large variability of RE, varying between positive and negative values, suggests that larger eddies (with time scales larger than 30 min) contribute to both positive and negative transport. This fact, by itself, is an indication that averaging times greater than 45 minutes are accounting for the effects of the submesoscale (non-turbulent) motions [2][3]. Therefore, 45 minutes can be approximately the timescale of the spectral gap.

- Thanks for the valuable insight. We have incorporated this point in the revision document (lines: 355 to 358) as:

"A high variability in RE for time scales larger than 45 min indicate the effects of sub mesoscale (non-turbulent) motions. Hence, 45 min average period can be considered as optimal in isolating the turbulence components for use with flux representation".

13) Figures 5a) and 5b): Just a suggestion: normalize the ogive by the maximum value of each curve (integrated up to the lowest frequency). This parameter indicates the fractional contribution of each frequency to the total cumulative energy.

- Since our interest is restricted to obtain the optimal time-averaging period (inflection point on Ogive plot), we have not normalized.
- However, we will consider this suggestion in our future studies.

14) Lines 404-405 and Figure 8): "This conclude that, the need for optimal averaging period is more crucial in estimating WUE fluxes rather than individual carbon and water fluxes." This is what can be observed in Figure 8. However, it is not clear why the large linear correlation between the fluxes with different averaging times is not observed on the WUE chart. According to May appointments 9) and 11), a large variability in RE is observed. By definition of the WUE, we have to consider the ratio of two fluxes and their respective RE. My first guess is that this loss of linear correlation is associated with the difference in RE between the fluxes of $CO_2$ e $H_2O$.

- Agreed with the reviewer. Its is observed from the figure 8, the need for optimal time averaging period is more crucial in estimating WUE rather than its individual fluxes i.e. carbon and water.
- Rather than poor correlations, this is an interesting finding. Though carbon and water fluxes are strongly correlated individually, their ratio term (WUE fluxes) is poorly correlated between any two averaging periods.
- For any two averaging periods (refer link: https://zenodo.org/badge/latestdoi/528291820), variation in carbon sink and water vapour are low, whereas variation in WUE is high.
- For example, the following table provides the carbon, water, and WUE fluxes at 15 min and 45 min averaging periods for some selective dates.
- Observe the huge differences in WUE (5.42 to 20.21 %) in comparison to the differences in carbon (0.14 to 10.37 %) and water (0.25 to 8.60 %) fluxes.
- Hence, variation in WUE between any two time-averages is much larger than variation in carbon or water fluxes.

- For this reason, we are highlighting that, choice of optimal time-averaging is crucial in WUE analysis, rather than carbon or water flux analysis.

| Day | CO$_2$ [$\mu mol^{+1}$ s$^{-1}$ m$^{-2}$] | | Deviation in CO$_2$ Flux [%] | H$_2$O [$\mu mol^{+1}$ s$^{-1}$ m$^{-2}$] | | Deviation in H$_2$O Flux [%] | WUE [$\mu mol$ mmol$^{-1}$] | | Deviation in WUE Flux [%] |
|---|---|---|---|---|---|---|---|---|---|
| | 15 min | 45 min | | 15 min | 45 min | | 15 min | 45 min | |
| 1 | -0.3719 | -0.37144 | 0.1415 | 1.4518 | 1.4555 | 0.2571 | 0.3714 | 0.29669 | 20.1315 |
| 2 | -0.7048 | -0.63173 | 10.3755 | 1.1326 | 1.1164 | 1.4354 | 1.3640 | 1.08831 | 20.2168 |
| 3 | -1.4137 | -1.4207 | -0.4905 | 1.0586 | 1.1497 | 8.6037 | 1.7367 | 1.48193 | 14.6742 |
| 4 | -0.7963 | -0.7929 | 0.4221 | 0.9210 | 0.9152 | 0.6357 | 1.4913 | 1.41033 | 5.4297 |

- Also note that, the correlation strengths were plotted considering the entire crop cycle dataset, rather than individual growth stages.
- These causes are explained in the revised manuscript (lines: 448 to 451) as follows:

"However, a poor linear association in WUE fluxes was observed between any two averaging periods, which is attributed to a larger variation in individual WUE fluxes between averaging periods. However, the corresponding individual carbon and water fluxes have recorded low variations between time averages".

**III- Thecnical corretions:**

1) Whole text: "min" is the symbol for the time unit "minute". Therefore, the form "min." with a dot is unusual.
- Thank you for the suggestion.
- We replaced "min." with "min" to represent "minute" in the entire manuscript.

2) Line 130: Change "0C" to "oC".
- Thank you for the suggestion. Correction applied.

3) Line 244: Change "Where" by "where".
- Thank you for the suggestion. Correction applied.

4) Figure 3 a) and b): If possible, write the flux units in the correct notation.

- Thank you for the suggestion. Units are properly represented in Figure 3a and 3b.

5) Figure 6: Legend – change "(red)" by "(dotted)".

- Thank you for the suggestion. Figure 6 legend got changed.

6) Figure 4 and 7: Change time units from "[Min]" to "[min]".

- Thank you for the suggestion. We modified the time units from "[Min]" to "[min]" in Figure 4 and 7.

7) Line 488 and 581: To correct these references, there are typing errors.

- Thank you for the suggestion. References got modified.

---

## Referee Report (RR1)

The authors have carefully considered the reviewer's comments and the manuscript has significantly improved. I suggest to pay attention on the few comments below, in particular the comment 7.

Detailed comments

1. L. 16-17. The canopy heat storage is not the only reason for the departure of the EBR from unity. The current study was investigating the impact of low-frequency flux loss as one of the reasons; there are also potential other reasons such as impact of advection and others. In general, the incomplete energy balance closure is a well known problem (as acknowledged also by the authors in the manuscript) and is not definitely uniquely attributed to the heat storage.

2. L. 22, the reported values of the WUE during different periods are defined by the average number +- variation range. It is not self-explaining what these values after +- mean. I raised the same question earlier in the first revision in relation to reported climatic average temperatures (section 2.1). The authors responded that it is a common practice in statistics to represent the mean +- standard deviation. I argue that it is not uniquely clear: it is a common practice to represent experimental results mean+-error, which frequently means the standard error of the mean. Or, it could be also the confidence intervals of the mean at some significance level. Therefore, to be clear, I always prefer that it is specified what variation range is presented.

3. L. 135, presumably the +- values represent here the inter-annual variability (standard deviation) of the summer and winter seasons mean temperatures over a range of years. Again, it is not self-explaining. Also, the +- values do not provide here additional information and could be omitted.

4. L. 168 and the averaging periods: it was explained that the co-ordinate rotation was performed over the same time interval as the averaging period. Presumably 1 minute is in general too short time period to define a "stable" co-ordinate system. The authors should admit that such a short averaging period introduces significant random uncertainty due also due to co-ordinate rotation bound to the same period.

5. L. 326-328, the sentence is difficult to understand. Could it be something like "We did not observe variation of optimal averaging time with wind speed and direction, hence".

6. Fig. 8: Please specify in the figure title that the circle size represents the correlation magnitude. And the colour scale the sign? Did you observe also negative correlations? If not, it would be more clear to represent the colour scale from 0 to 1. If there are negative correlations on the average, then how do you interpret those?

7. L. 448-450: if I understand the fig 8c correctly, then it is not true that a poor correlation was observed between any two averaging periods. The correlation between the periods 15 min and 45 mins looks close to 1. Also, the correlation between 1 min and 30 min averaging is fairly high. It is natural that the correlation of WUE (which is the ratio of the two fluxes with both having their random uncertainty) values for different averaging periods is lower. However, it is counter intuitive that the correlation between 15 min and 45 min averaging periods is high, but for 15 mins vs 30 mins (which are more close averaging periods) is completely lost. If such behaviour results from uncertainty in single WUE values (or possibly correlations being impacted by "outliers"), then Fig. 8c does not serve

as useful information providing insights into WUE dynamics and should be omitted.

---

## Referee Report (RR2)

Most of my questions were answered by the authors in the first round of reviews. However, I have some suggestions and minor corrections that can improve the manuscript.

**I - General comments:**

1) Beside the main objective of obtaining the optimal averaging time period, the authors also evaluated $CO_2$ and water vapor fluxes for the different growth stages. I think this result is also relevant. There is no discussion relating the average flux intensity to the growth stages (figures 3a and 3b). Are these observed flux differences in different growth stages spected? Are there studies showing similar results that can be cited? This discussion can be included in the present manuscript if the authors deem it relevant.

2) Line 316: "particularly with dough and maturity stages due to ignorance of canopy heat storage." and line 322: "Low EBR during the crop cycle can also be attributed to the ignorance of energy transport associated with large eddies from landscape heterogeneity.". The authors must cite studies that support those hypotheses.

**II- Specific comments:**

1) Line 120: "…compute optimal averaging period to simulate carbon and water (hence, WUE) fluxes of Maize crop,". In my understanding, "to simulate" should be changed to "to evaluate" or "to calculate". No simulation is mentioned in the manuscript.

2) Check equation 10 ($R^2$): The current equation provides a dimensional parameter.

3) Figure 1 (and throughout the manuscript): you should use a standard abreviation to the minute unit ("min" not "Min").

4) Figure 8). Caption referes to subplots a, b and c, but those identificantion are not shown in the figure.

5) As I understood, the correlation chart shows the correlation evaluated using the series of daily averaged fluxes calculated using different average windows. Am I right? The methodology used to evaluate these correlations should be clarified in the manuscript .

6) Line 448: What is the parameter rho ($\rho$)? It is not defined in the text. Is it associated with the $R^2$ or the r parameter?

---

## Author Response (AR2)

Manuscript Number: AMT-2023-253

Manuscript Title: Role of time-averaging of eddy-covariance fluxes on water use efficiency dynamics of Maize crop.

**AUTHORS' RESPONSES TO REVIEWER – 2 COMMENTS**

The authors have carefully considered the reviewer's comments and the manuscript has significantly improved. I suggest to pay attention on the few comments below, in particular the comment 7.

- Thanks to the reviewer for valuable comments and appreciation.
- We have addressed all the comments and provided our responses in a detailed manner.

**General comments**

Detailed comments

1. 1. L. 16-17. The canopy heat storage is not the only reason for the departure of the EBR from unity. The current study was investigating the impact of low-frequency flux loss as one of the reasons; there are also potential other reasons such as impact of advection and others. In general, the incomplete energy balance closure is a well known problem (as acknowledged also by the authors in the manuscript) and is not definitely uniquely attributed to the heat storage.

   - Agreed that, apart from canopy heat storage, there a few other possible reasons for departure of EBR from unity as listed below:
        i) Ignoring canopy heat storage terms
        ii) Advection due to movement of water (evaporation, rainfall, etc.) through the atmosphere in the ecosystem.
        iii) Low-frequency circulations (large eddies) losses
        iv) Inadequate time-averaging techniques
        v) Improper selection of suitable measuring locations in heterogeneous flows
        vi) Instrumentation and systematic errors
   - In this study, we ensured that the instrumentation and systematic errors are negligible and measurements are performed in homogeneous croplands. So, we omitted these factors responsible for EBR departure.
   - Regarding advection, we segregated the data into advection ($H < 0$ and $LE > 0$) and non-advection ($H > 0$ and $LE < 0$) components, and obtained a weaker correlation

between temperature and relative humidity. Hence, we conclude that advection is not a major source for EBR departure (Rahman et al., 2019). This is also due to the fact that, the study site is not surrounded by water reservoirs or other sources.

- Hence, we only investigated the other major sources like canopy heat storage, low frequency flux loss, inadequate time-averages on EBC closure in this study.

- However, to give more clarity, we modified the given statement in the revised manuscript as follows (Line: 16 to 18):

  "A clear departure of EBR from unity was observed during dough and maturity stages of the crop due to ignorance of canopy heat storage, low frequency flux losses and inadequate averaging period".

2. L. 22, the reported values of the WUE during different periods are defined by the average number +- variation range. It is not self-explaining what these values after +- mean. I raised the same question earlier in the first revision in relation to reported climatic average temperatures (section 2.1). The authors responded that it is a common practice in statistics to represent the mean +- standard deviation. I argue that it is not uniquely clear: it is a common practice to represent experimental results mean+-error, which frequently means the standard error of the mean. Or, it could be also the confidence intervals of the mean at some significance level. Therefore, to be clear, I always prefer that it is specified what variation range is presented.

- We sincerely apologize for the confusion created in this regard.

- When representing the variation in data series, we use μ±σ (mean ± standard deviation), as this explains the most of the data variation.

- When conveying the precision of the sample mean, we use μ±SE (mean ± standard error), as this provides an estimate of the population mean.

- It should be noted that, both SD and SE are inter-related. $SE = \frac{SD}{\sqrt{n}}$

- However, to avoid further confusion:
  1. We used μ±σ when denoting data range / dispersion in a given series
  2. We used mean (μ) alone, when representing the central tendency of data series
  3. When used for the first time, we clearly expanded the terms, μ (or) μ±σ in the revision

3. L. 135, presumably the +- values represent here the inter-annual variability (standard deviation) of the summer and winter seasons mean temperatures over a range of years. Again, it is not self-explaining. Also, the +- values do not provide here additional information and could be omitted.

- Again, sorry for any confusion.
- We have specified the terms μ and σ (line: 135) to avoid confusion.
- SD represented here ($\pm 2$ °C) gives an indication that, most of the data (high and low temperatures) is concentrated around the mean, and the variability is low.

4. L. 168 and the averaging periods: it was explained that the co-ordinate rotation was performed over the same time interval as the averaging period. Presumably 1 minute is in general too short time period to define a "stable" co-ordinate system. The authors should admit that such a short averaging period introduces significant random uncertainty due also due to co-ordinate rotation bound to the same period.

- Agree with the reviewer that 1 min is too short to define stable co-ordinate rotation, hence introduces additional random uncertainty.
- We also obtained similar results, as 1 min time averaged data has a large scatter (randomness) which is evident from the inset of Figure 1.
- We incorporated the referee's suggestion in the revised manuscript (line: 292 to 296) as follows:

"The variation is rough at lower averaging periods due to a high sample size (n = 10859 at T = 1 min) and is gradually smoothened towards higher averaging periods (n = 811 at T = 120 min). The shorter averaging periods has introduced random uncertainty in the datasets during co-ordinate rotation correction."

5. L. 326-328, the sentence is difficult to understand. Could it be something like "We did not observe variation of optimal averaging time with wind speed and direction, hence".

- Agreed. This sentence is modified in the revised manuscript (line: 329 to 330) as follows:

"We did not observe variations in optimal averaging time due to changes in wind speed and direction, hence meteorological conditions were not analysed in this study."

6. Fig. 8: Please specify in the figure title that the circle size represents the correlation magnitude. And the colour scale the sign? Did you observe also negative correlations? If not, it would be more clear to represent the colour scale from 0 to 1. If there are negative correlations on the average, then how do you interpret those?

- Agree with the reviewer, and accordingly revised the figure title.

- Yes, we observed a negative correlation only with WUE, but not with $CO_2$ or $H_2O$ fluxes (Figure 8c).

- Negative correlations are mainly due to the inverse relations between two datasets, as $CO_2$ acts as either sink (daytime) or source (night-time), and $H_2O$ fluxes always acts as source. WUE, being the ration between the two fluxes has resulted in either positive or negative variations considering different averaging periods.

- For ease with understanding, the dynamics of carbon (top row), water (middle row) and WUE (bottom row) for 3 averaging periods (15-, 30-, 1md 45-min) are given below.

[Figure]

- Observe similarities in carbon (or) water fluxes between different averaging periods, as well as dissimilarities in WUE fluxes between different averaging periods. This has resulted in all-together a different set of correlations.
- Additionally Figure 8c is not plotted as per crop growth stages, hence stage-wise variations were not included in the final values.

7. L. 448-450: if I understand the fig 8c correctly, then it is not true that a poor correlation was observed between any two averaging periods. The correlation between the periods 15 min and 45 mins looks close to 1. Also, the correlation between 1 min and 30 min averaging is fairly high. It is natural that the correlation of WUE (which is the ratio of the two fluxes with both having their random uncertainty) values for different averaging periods is lower. However, it is counter intuitive that the correlation between 15 min and 45 min averaging periods is high, but for 15 mins vs 30 mins (which are more close averaging periods) is completely lost. If such behaviour results from uncertainty in single WUE values (or possibly correlations being impacted by "outliers"), then Fig. 8c does not serve as useful information providing insights into WUE dynamics and should be omitted.

- This is mainly due to the behaviour of fluxes at different averaging periods. The above figure presents the dynamics of carbon, water, and WUE fluxes at 15-, 30-, and 45-min averaging periods.
- A close inspection of these figures concludes that:
  - Variation in carbon and water fluxes is similar (both trend and magnitude) among all three averaging periods. Hence, we achieved high correlation with carbon (Figure 8a) and water (Figure 8b) fluxes between any two averaging periods.
  - However, variation in WUE fluxes is similar in magnitude between 15- and 45-minute averaging periods only, but not between 15- and 30-minute averaging periods.
  - While we could not certainly ascertain the reasons, this conclude that conventional 30-min averaging may not be appropriate throughout the growth period in representing the WUE fluxes.

- With this, we think that the Figure 8c provides an important insights and information on WUE dynamics at different time averages.

Manuscript Number: AMT-2023-253

Manuscript Title: Role of time-averaging of eddy-covariance fluxes on water use efficiency dynamics of Maize crop.

**AUTHORS' RESPONSES TO REVIEWER – 3 COMMENTS**

Most of my questions were answered by the authors in the first round of reviews. However, I have some suggestions and minor corrections that can improve the manuscript.

- Thanks to the reviewer for providing valuable suggestions and comments.
- We have addressed all of your comments in a detailed manner.

**I - General comments:**

1) Beside the main objective of obtaining the optimal averaging time period, the authors also evaluated $CO_2$ and water vapor fluxes for the different growth stages. I think this result is also relevant. There is no discussion relating the average flux intensity to the growth stages (figures 3a and 3b). Are these observed flux differences in different growth stages spected? Are there studies showing similar results that can be cited? This discussion can be included in the present manuscript if the authors deem it relevant.

- We observed that averaging period has little to no influence on mean carbon and water fluxes, averaged over each growth stage. This is particularly true with 6th leaf and silking stages. However, maturity and dough stages are exceptional, particularly with 45-min averaging period.

- Variation in flux intensity with growth stage is mentioned in lines: 344 to 346 of the revision.

- Following reviewer suggestion, averaging flux intensity variation with growth period is provided in the revision (lines: 346 to 352) along with a discussion and literature support.

"From the mean $CO_2$ and $H_2O$ flux dynamics, it is observed that the drip irrigated Maize crop is acting as a carbon sink in the entire crop season especially in the latter stages of the crop i.e. maturity stage with a mean of 15.44 $\mu$mol m$^{-2}$s$^{-1}$. This is clearly evident from the increasing trend of LAI and plant height during the crop season. Such an increase is highlighted by previous studies of Guo et al., 2021. At the same time, mean $H_2O$ fluxes were increased towards the end of crop growing season due to increased crop water demand."

2) Line 316: "particularly with dough and maturity stages due to ignorance of canopy heat storage." and line 322: "Low EBR during the crop cycle can also be attributed to the ignorance of energy transport associated with large eddies from landscape heterogeneity.". The authors must cite studies that support those hypotheses.

- Agree with the reviewer, and we have added the citations that support our hypothesis about the low EBR (line: 327 of revision).

**II- Specific comments:**

1) Line 120: "…compute optimal averaging period to simulate carbon and water (hence, WUE) fluxes of Maize crop,". In my understanding, "to simulate" should be changed to "to evaluate" or "to calculate". No simulation is mentioned in the manuscript.

- Agreed, and changed as below:

  "Identify optimal averaging period to evaluate carbon and water (hence, WUE) fluxes of Maize crop."

2) Check equation 10 ($R^2$): The current equation provides a dimensional parameter.

- It's a typo. The equation is modified in the revised manuscript as follows:

$$R^2 = \left\{ \frac{\sum_{i=1}^{n}\left[(R_n - G)_i - \overline{(R_n - G)}\right]\left[(H + LE)_i - \overline{(H + LE)}\right]}{\sqrt{\sum\left[(R_n - G)_i - \overline{(R_n - G)}\right]^2\left[(H + LE)_i - \overline{(H + LE)}\right]^2}} \right\}^2$$

3) Figure 1 (and throughout the manuscript): you should use a standard abreviation to the minute unit ("min" not "Min").

- We have changed the abbreviation for minute in the Figure 1 and we followed a standard abbreviation of "min" for representing the minute unit in throughout the manuscript.

4) Figure 8). Caption referes to subplots a, b and c, but those identificantion are not shown in the figure.

- Sorry, we have provided the identifications for the subplots in Figure 8 in the revised manuscript.

5) As I understood, the correlation chart shows the correlation evaluated using the series of daily averaged fluxes calculated using different average windows. Am I right? The methodology used to evaluate these correlations should be clarified in the manuscript.

- Sorry to the reviewer for any confusion.
- Yes, the correlation chart shows the correlation evaluated using the series of daily averaged fluxes calculated using different average windows.
- We modified the given sentence in the revised manuscript (line: 449 to 450) as follows: "Correlation charts showing the linear association considering daily means of carbon, water, and WUE fluxes at different averaging periods is represented in Figure 8."

6) Line 448: What is the parameter rho ($\rho$)? It is not defined in the text. Is it associated with the $R^2$ or the r parameter?

- Sorry to the reviewer for the confusion.
- This is associated with the parameter "r" and the is modified in the revised manuscript.